# FUBP1 and FUBP2 enforce distinct epigenetic setpoints for MYC expression in primary single murine cells

Ying Zheng[1], Wendy Dubois[2], Craig Benham[3], Eric Batchelor[4] & David Levens[1✉]

Physiologically, MYC levels must be precisely set to faithfully amplify the transcriptome, but in cancer MYC is quantitatively misregulated. Here, we study the variation of MYC amongst single primary cells (B-cells and murine embryonic fibroblasts, MEFs) for the repercussions of variable cellular MYC-levels and setpoints. Because FUBPs have been proposed to be molecular "cruise controls" that constrain MYC expression, their role in determining basal or activated MYC-levels was also examined. Growing cells remember low and high-MYC setpoints through multiple cell divisions and are limited by the same expression ceiling even after modest MYC-activation. High MYC MEFs are enriched for mRNAs regulating inflammation and immunity. After strong stimulation, many cells break through the ceiling and intensify MYC expression. Lacking FUBPs, unstimulated MEFs express levels otherwise attained only with stimulation and sponsor *MYC* chromatin changes, revealed by chromatin marks. Thus, the FUBPs enforce epigenetic setpoints that restrict MYC expression.

[1] Lab of Pathology, National Cancer Institutes, Bethesda, MD, USA. [2] Lab of Cancer Biology and Genetics, Center for Cancer Research, National Cancer Institutes, Bethesda, MD, USA. [3] Biomedical Engineering, University of California, Davis, CA, USA. [4] Masonic Cancer Center and Department of Integrative Biology and Physiology, University of Minnesota, Minneapolis, MN, USA. ✉email: levensd@mail.nih.gov

From stem cells and proliferation to senescence and cell death, MYC is involved in almost every key decision for every cell[1–7]. MYC is an amplifier of almost every active gene and it becomes an oncogene when abnormally upregulated[8–11]. On the other hand, although the knockout (KO) of *Myc* is embryonic lethal in mice and *Myc*-KO cells have only rarely been obtained (less than a handful of *Myc*-less cell lines have been established)[12,13], very low levels of basal MYC are sufficient to sustain viability[13–16]. *MYC*-haploinsufficient cells are not notably abnormal other than pro-liferating at a somewhat reduced rate. Haploinsufficient mice dis-play no pathology and show increased life span[15,17–22]. Despite this apparent indifference to normal levels, constraining MYC to a physiological range is crucial for normal development and phy-siology. In response to a variety of intracellular and extracellular signals and pathological stresses, MYC levels rapidly elevate to enable transcriptional amplification of highly induced genes, as the cell races to confront the challenging insult[23,24]. Such high-MYC output in cancer may also be enforced by a panoply of pathologi-cally activated super-enhancers[14,25–28].

*Myc* is embedded in a gene desert and that is studded with *cis*-elements receiving a plethora of extracellular and intracellular signals that must somehow be integrated at the promoter to set MYC expression levels[23,24,29]. Of the multiple regions that reg-ulate MYC expression, we have been interested in the far upstream element (FUSE), which is 1.7 kb upstream of the major P2 transcription start site of human *MYC* gene and binds FUBP1[23,29–36]. The FUSE-binding protein (FUBP) family has three members in human and mouse—FUBP1, FUBP2/KHSRP, and FUBP3[23,29–35,37,38]—and only one in *Drosophila*—Psi[31,32,37,39]. FUBP1 is both prooncogenic and a tumor suppressor[37,40,41]. The FUBPs all contain four repeated hnRNPK homology motifs in their central regions. Their central regions bind sequence selectively to either single-stranded DNA or RNA and the FUBPs have been reported to interact with various DNA or RNA targets and to participate in transcription, splicing, mRNA degradation, RNA transport, and translational regulation[29,31,32,34–38,42,43]. After FUSE melts in response to the torsional stress caused by dynamic supercoiling emanating from the *Myc* promoter, FUBP1 binds and loops to the promoter, interacting with TFIIH through the C-terminal domain of FUBP1 to activate transcription[44,45]. The FUBP1-interacting repressor (FIR) may then be recruited by FUBP1 to inhibit the helicase activity of TFIIH, attenuating FUBP1 upregulation of *Myc* transcription[43,45–47]. Thus, the FUSE-FUBP-FIR system has been proposed to comprise a cruise control constraining MYC expression[29,31,32,45,48–50] by increasing MYC expression in some circumstances and decreasing it in others. Accordingly, when *Fubp1* is knocked out in mice, both cell-to-cell and embryo-to-embryo Myc RNA levels fluctuate[51].

Intercellular expression variation occurs between different tis-sues and between individual members of a single-cell type. Almost all types of cells, including stem cells, murine embryonic fibroblasts (MEFs), and cancer cells, are not a homogenous population. They are all heterogeneous at the molecular level. The degree to which this heterogeneity reflects transient stochastic fluctuations (intrinsic noise) vs. biologically distinct, perhaps epigenetic states (extrinsic noise) is not fully resolved[52]. Such cell-to-cell molecular and functional variation of MYC may be expected to amplify in turn the quantitative or even qualitative diversity of responses to stimuli. Understanding the sources of this variation may provide insights into how cells respond indi-vidually or in coordinated groups to physiological and patholo-gical challenges.

Variation in basal MYC levels is well-tolerated in resting and steady-state tissues, yet there are multiple molecular mechanisms that control its expression across its full dynamic range. We set out to examine the precision and variability of MYC regulation and to study the cellular consequences of variation in MYC set-points both for steady-state growth and in response to cytokine stimulation in primary cells. Primary MEFs and primary naive B-cells were studied to observe *Myc* regulation in the absence of oncogenic stress. The induction of MYC in B-cells has been well-characterized in vivo and in vitro, and the use of wild-type (WT) vs. *FUBP1*$^{-/-}$ MEFs, with or without *Fubp2* knockdown (KD), provided a simple system to interrogate the roles of these proteins in setting MYC levels, in managing expression noise, and in upregulating MYC in response to cytokine signaling. We found that there is a two-stage upregulation of *Myc*; in response to low-intensity signals and stress, although MYC is broadly upregulated, it respects a physiological upper limit. With intense signaling, a significant portion of cells breaks the ceiling, with some cells expressing very high-MYC levels. As elevated MYC is seen in almost all types of cancers, neoplastic cells must defeat whatever safeguards enforce physiological MYC levels. These studies may help inform strategies to restore proper regulation on deregulated MYC.

## Results

**Characterization of Myc mRNA and protein expression in single cells**. MYC is predominantly a nuclear protein (Fig. 1a, c). To ensure that MYC levels could be accurately monitored in individual cells microscopically, primary MEFs and primary naive B-cells were fixed, their nuclei segmented based on DNA staining, and then immune-stained for MYC protein. Nuclear size and the mean nuclear fluorescence intensity of MYC were measured in single MEFs (Fig. 1a, b) or naive B-cells (Fig. 1c, d). The total amount of MYC protein was calculated as the product of mean intensity and nuclear size. To validate this measurement, naive B-cells were stimulated with lipopolysaccharide (LPS) and inter-leukin (IL)-4, and split into two portions: one for flow cytometric analysis and the other for quantitative immunofluorescence microscopy. Comparable results and distributions were obtained with each method, validating both approaches (Fig. 1e). For the determination of Myc RNA levels after in situ hybridization with fluorescent probes for Myc RNA, the probe intensity was measured at the level of individual cells and the distribution of MYC across the full population of MEFs was determined (Fig. 1f, g). Total Myc mRNA intensity was calculated as the product of mean intensity and cell size. The density profiles of MYC protein or mRNA levels of primary MEFs and naive B-cells were plotted (Fig. 1b, d, g); both protein and RNA show similar bell shapes with a long tail extending toward a high expression that approximated a log-normal distribution (Supplementary Fig. 1).

**Two-stage regulation of MYC expression**. To characterize the response of the *Myc* gene to extracellular signals at both the population and single-cell levels, naive B-cells or MEFs made quiescent by serum starvation were stimulated with different concentrations of IL-4 or fibroblast growth factor (FGF), respectively (Fig. 2). The concentrations of growth factors and cytokines were empirically defined to deliver partial or maximal induction of MYC. These concentrations also approximate levels seen under normal or stressed/pathological concentrations in vivo in various systems. Cumulative distribution functions (CDFs) were used to capture both the single-cell and population prop-erties in a single format amenable to statistical comparison (Fig. 2a). Upon stimulation of B-cells with low concentrations of IL-4, the MYC distribution shifted rightward, without changing the peak values of the MYC expression (Fig. 2b). Thus, although the overall population was MYC responsive, it respected the same

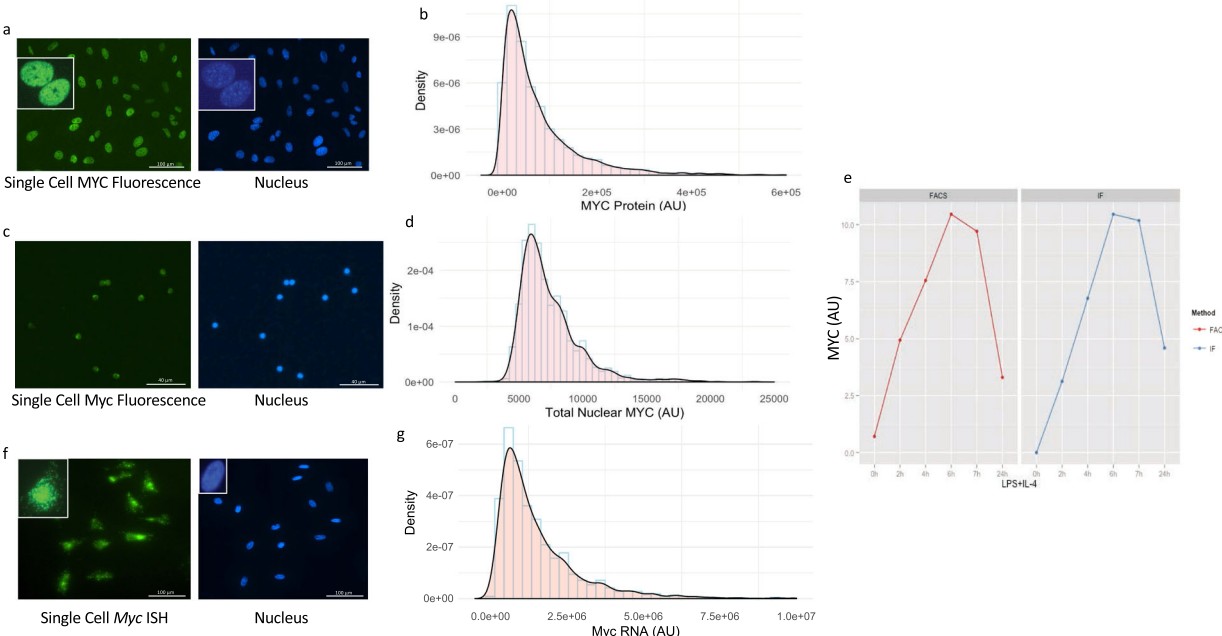

**Fig. 1 Distribution of MYC expression in single cells. a**, **b** MYC protein expression in single MEF by immunostaining. **a** An immunostaining image of MYC protein in single MEFs, showing that MYC is primarily localized in the nucleus of MEFs. **b** Histogram (light blue) and density plot (black) of MYC total nuclear protein level determined by immunofluorescence. **c**, **d** MYC protein expression in single B-cells by immunostaining. **c** An immunostaining image of MYC protein in single B-cells, showing that MYC is primarily localized in the nucleus. **d** Histogram (light blue) and density plot (black) of MYC total nuclear protein level in single naive B-cells are shown. AU: arbitrary unit. **e** Validation of quantitative immunofluorescence. Primary naive B-cells were stimulated with 25 µg/ml of LPS and 2.5 ng/ml of IL-4, and split into two parts for flow cytometry analysis (FACS) and quantitative immunofluorescence analysis (IF). Comparable results were obtained. **f**, **g** MYC mRNA expression in single MEF by in situ hybridization. **f** An in situ hybridization image of MYC mRNA in single MEFs. **g** Histogram (light blue) and density plot (black) of Myc RNA level determined by RNA in situ hybridization. AU: arbitrary unit.

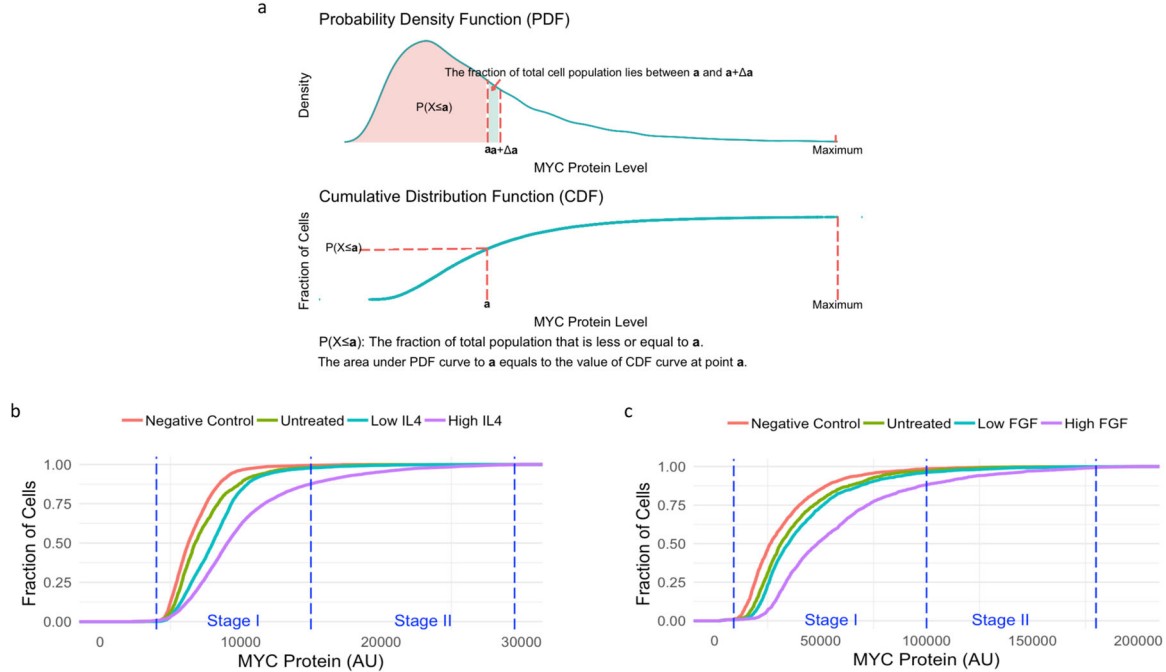

**Fig. 2 MYC expression has two stages: low (I) and high (II) stages. a** A diagram illustrating the principles and biological implications of the PDF and the CDF plots used in this study. A random log-normal distribution data set were used as an example. **b**, **c** The empirical CDF plot of MYC protein level in naive and IL-4-stimulated B-cells (**b**), and in steady-state and FGF-stimulated MEFs (**c**) are shown. Low IL-4: 5 ng/ml; high IL-4: 50 ng/ml. Low FGF: 1 ng/ml; high FGF: 4 ng/ml. AU: arbitrary unit. Negative control: nonspecific IgG isotype control. Experiments were repeated with similar results ($n = 3$ independent experiments).

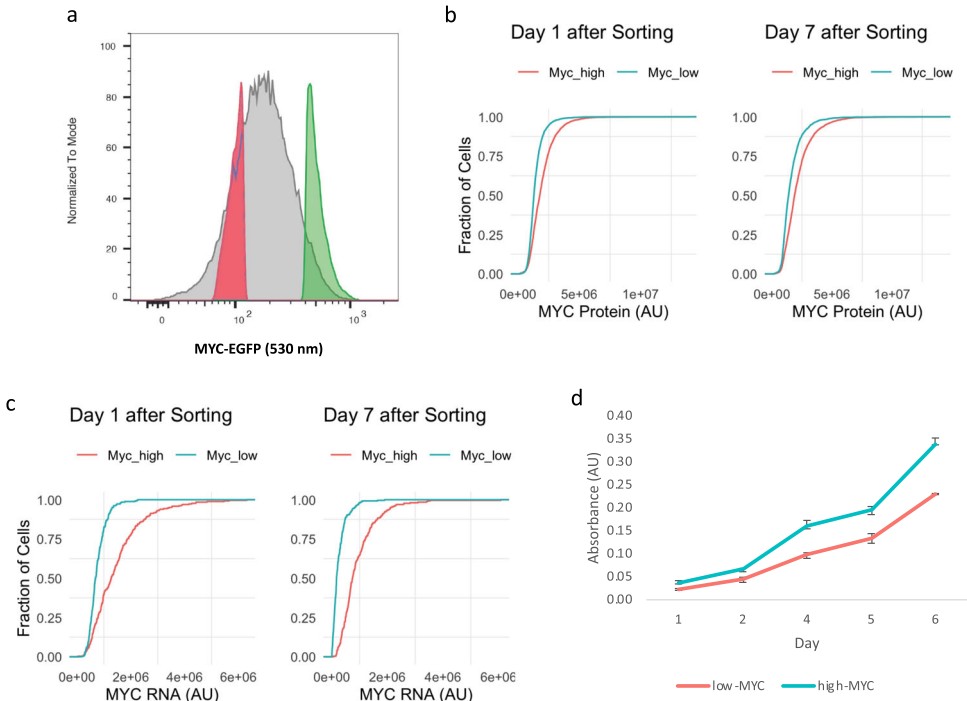

**Fig. 3 MYC setpoints are precisely set and fixed in individual cells. a** Sorting scheme for low- and high-MYC cells by MYC-GFP intensity. **b**, **c** The empirical CDF plot of MYC protein (**b**) and mRNA (**c**) expression in cells at day 1 (left) or 7 (right) after sorting. **d** The proliferation of low- and high-MYC cells determined by WST-1 assay. Experiments were performed in triplicate.

expression ceiling as did the unstimulated cells. With high concentrations of IL-4, the CDF shifted further to the right and breached the expression ceiling of unstimulated cells extending to much higher levels of MYC. The MYC CDF curve from MEFs treated with low levels of FGF also shifted rightward and paralleled that from MEFs growing at steady-state with both curves converging to a common upper limit (Fig. 2c). With high FGF, the CDF shifted dramatically rightward with ~20% of cells expressing two to three times the steady-state mean MYC levels. These data suggest that there are two stages of MYC expression as follows: low (Stage I) and high (Stage II). Stage I (which we operationally define as physiological) occurs at a steady state or upon mild stimulation—in Stage I, although *Myc* is inducible, overall MYC expression is strictly bounded. Stage II (which we operationally define as stressed/pathological) occurs upon intense stimulation with concentrations of growth factors or cytokines that maximize MYC expression, allowing a population of cells to breach the Stage I ceiling and express much higher MYC levels. Plotting MYC distribution as a probability density function of B-cells or MEFs treated with IL-4 or FGF, respectively (Supplementary Fig. 2a, b), confirmed that upon weak stimulation, mean levels shifted to modestly higher values in stage I, but that stage II was not populated without intense stimulation.

**MYC setpoints are precisely set and fixed in individual cells**. Within stage I, the variation in MYC levels between individual cells could reflect stochastic fluctuation with sampling from a single distribution or, alternatively, the superposition of closely nested distributions, each with slightly shifted mean MYC setpoints.

To test whether there is a common setpoint for MYC expression among individual MEFs, fluorescence-activated cell sorting was used to collect a set of low- and a set of high-MYC-expressing cells from an apparently unimodal distribution of fluorescent MEFs (Fig. 3a). These MEFs were homozygous for a

fully functional chimeric *Myc-Egfp* gene expressed at the endogenous *Myc* locus. Mice homozygous for this fusion allele are phenotypically normal and breed properly. This in-frame chimera of enhanced green fluorescent protein (GFP) at the C terminus of MYC preserves all known features that regulate Myc RNA and protein expression and turnover. We next cultured the low- and high-MYC cells for 1 vs. 7 days (at least two mitotic cycles) and then examined their distributions of MYC protein and mRNA. Myc mRNA and protein are relatively unstable (mRNA half-life is 10–20 min[53] and protein half-life is about 20 min[54]). Surprisingly, both the low- and high-MYC cells still preserved their respective MYC expression levels (Fig. 3b, c). As expected, the high-MYC group doubled slightly faster and were slightly more metabolically active than the low-MYC group (Fig. 3d). These results suggest that MYC setpoints can be finely tuned and transmitted to daughter cells. After 1 week of culture, although the mean MYC levels in the low-MYC cells drifted slightly upward, their mode levels crept downward (Supplementary Fig. 3). The opposite shifts in the mode vs. the mean may be explained by the outgrowth of a subpopulation of cells extending the rightward tail toward higher MYC expression. Whether this subpopulation arises from setpoint reprogramming or the re-emergence of bona fide high-MYC-setpoint cells that factitiously happened to display low GFP fluorescence at the moment of sorting due to stochastic MYC fluctuation is not known.

We performed mRNA-seq on the low- and high-MYC cells, normalizing the count of transcripts to the count of spiked-in controls. As expected, the volcano plot shows a significant increase in gene expression in high-MYC cells compared to low-MYC cells and almost no downregulated genes as expected for a universal amplifier (Fig. 4a). The expression of *Myc*, *Fubp1*, and *Fubp2*/*Khsrp* in high-MYC cells is about twofolds of that in low-MYC cells (Supplementary Fig. 4). The mRNAs with the highest fold increase in high-MYC compared to low-MYC cells were highly enriched for genes involved in the inflammatory (9-fold

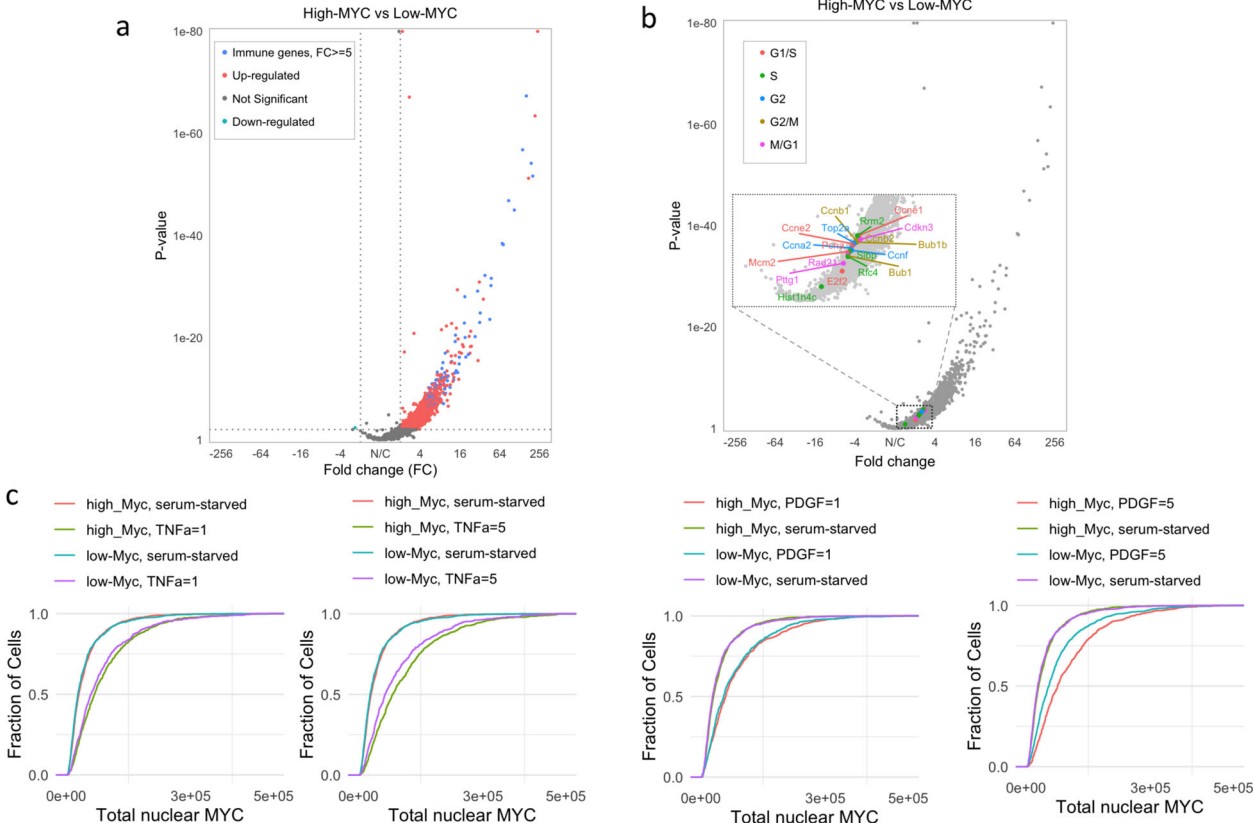

**Fig. 4 High-MYC cells are enriched for immune genes and are primed to express even higher MYC. a** The volcano plot of mRNA expression in high-MYC cells vs. low-MYC cells. In high-MYC cells, genes with the highest fold increase (FC ≥ 5) compared to low-MYC cells were enriched for inflammatory and immune response genes (GSEA analysis, FDR < 0.01), which are highlighted in blue. Each of the low-MYC and high-MYC RNA-seq samples has two technical replicates. **b** Compared with low-MYC cells, high-MYC cells are not enriched for well-defined cell cycle genes at specific stages. In the volcano plots of high-MYC and low-MYC cells, well-defined cell cycle genes at different stages are highlighted in different colors. **c** Effects of different doses of TNF-α or PDGF on the expression of MYC in low- or high-MYC cells. Unit of TNF-α or PDGF: ng/ml. AU: arbitrary unit. Experiments were repeated with similar results (n = 3 independent experiments).

enrichment, GO:0006954, false discovery rate (FDR) *q*-value < 0.01) and immune responses (5.8-fold enrichment, GO:0006955, FDR *q*-value < 0.01) and included various cytokines, chemokines, and complement (Supplementary Data 1). This suggests that high-MYC MEFs are primed to participate in immune and inflammatory processes, a major function of fibroblasts. The high-MYC cells were a bit larger than the low-MYC cells as expected[55,56]. This difference in size was maintained through several divisions of asynchronous culture and so could not be attributed to differences in the cell cycle distribution between the high- and low-MYC cells. Such asynchrony was confirmed by RNA sequencing (RNA-Seq), as the expression of mRNAs of genes expressed at well-defined points in the cell cycle[57] was indistinguishable between the high- vs. low-MYC populations (Fig. 4b), indicating that neither population was enriched for cells in particular phases of the cell cycle.

**High-MYC cells are primed to express even higher MYC.** After 7 days, low- and high-MYC cells were each serum-starved for 24 h, a condition known to arrest proliferation that puts cells into $G_0$ and depresses MYC levels. Upon serum starvation, both low- and high- MYC cells dropped to the same very low baseline levels of MYC (Fig. 4c). Then cells were re-stimulated with platelet-derived growth factor (PDGF) or tumor necrosis factor-α (TNFα) for 4 h and the level of re-expressed MYC was measured. A greater percentage of high-MYC cells were induced to higher MYC levels than were the low-MYC cells. Thus, following

quiescence, high-MYC cells are primed to support a greater MYC response. The results suggest that both the low- and high-MYC cells remember their setpoints through cell division in culture. The fact that high-MYC cells respond more vigorously to signals is expected, as MYC will amplify the outputs of those signals. The genes most expressed in the high-MYC cells encode mediators of inflammation and immunity suggesting that these cells may serve as sentinels that alert tissue to stress or pathological conditions.

To inquire whether the high vs. low-MYC cellular epigenetic memories derive from known epigenetic mechanisms, low- and high-MYC cells were treated with agents that alter epigenetic events, in an attempt to provoke a differential response to reprogram the MYC setpoints. When the low-MYC and high-MYC pools were treated with the DNA methylation inhibitor 5-azacytidine and/or the histone deacetylase (HDAC) inhibitor Trichostatin A, the MYC levels decreased proportionally in both pools, indicating that although DNA methylation and/or histone acetylation levels affected MYC levels, they were most likely not responsible for the memory of different MYC levels between cells (Supplementary Fig. 5). It is noteworthy that whereas for most genes HDAC inhibition and consequent hyperacetylation is associated with increased expression, in the case of *Myc*, expression falls.

**FUBP1 and FUBP2 police the MYC setpoint and enforce stage I in individual cells.** To examine the role of FUBPs in regulating MYC setpoints and their relationship to the stage I–II boundary,

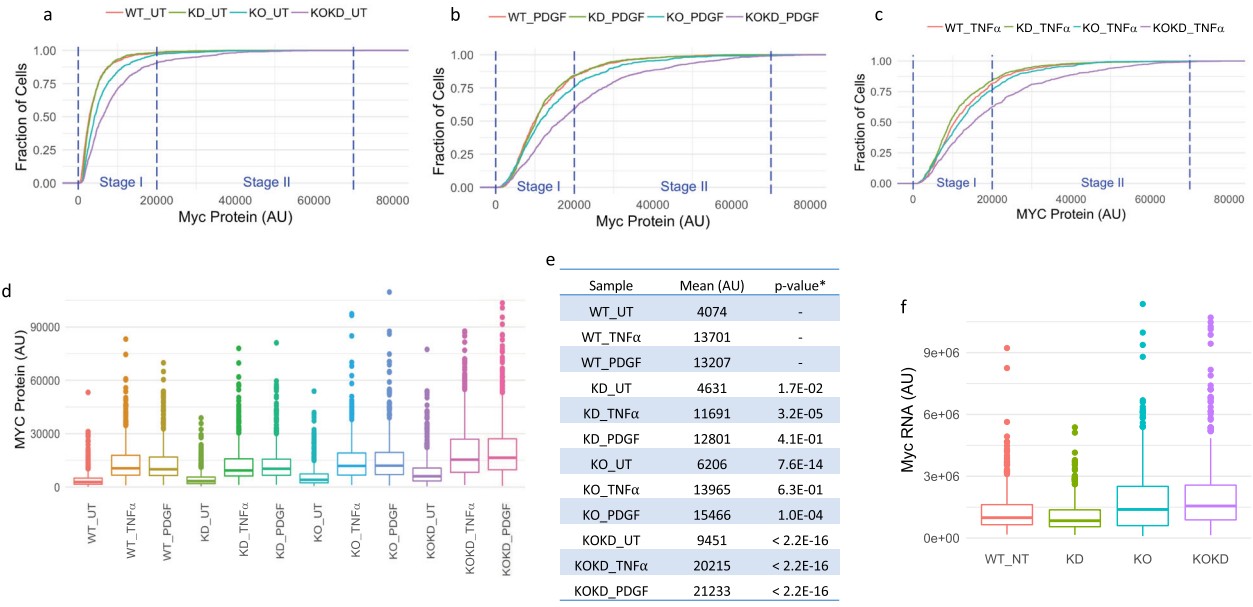

**Fig. 5 Loss of FUBP1 and FUBP2 breaks the boundaries between Stage I and II, and erases MYC setpoints in individual cells.** The empirical CDF plot of MYC protein level in untreated (UT) (**a**), PDGF-stimulated (**b**), or TNFα-stimulated (**c**) MEFs are shown. **d** Loss of FUBP1 and FUBP2 causes increased MYC protein levels and variation in untreated and stimulated cells. **e** The mean of MYC protein level of samples in **d**. *T-test assuming unequal variance, compared with the wild-type sample subjected to the same treatment. **f** Loss of FUBP1 and FUBP2 causes increased Myc mRNA levels and variation. TNFα or PDGF (2 ng/ml) was used for treatment. AU: arbitrary unit. Experiments were repeated with similar results ($n = 3$ independent experiments).

we examined MYC levels in single cells in four different populations of MEFs: (1) WT, 2) *Fubp1* KO[51], (3) *Fubp2* KD with small interfering RNA (siRNA) (KD), and (4) *Fubp1* KO plus *Fubp2* KD (KOKD). The KD of *Fubp2* in KD and KOKD cells was validated by immunoblotting (Supplementary Fig. 6a). These cells were studied after steady-state growth (Fig. 5a) or after serum starvation followed by stimulation with either PDGF (Fig. 5b), TNFα (Fig. 5c), or FGF (Supplementary Fig. 6b), and KOKD cells become inviable after ~5 days of culture. In MEFs grown at steady state, *Fubp2* KD does not perturb the MYC CDF. Although mean MYC levels deviated upwards upon *FUBP1* KO, the boundary separating stage I and stage II was not violated. The fact that increased mean MYC level in the KO was not accompanied by increased proliferation (Supplementary Fig. 6c) may be attributable to loss of FUBP1's direct impact on multiple cell cycle regulators[58]. The combined loss of *Fubp1* and *Fubp2* (KOKD) yielded exaggerated and erratic MYC levels that even at steady state cracked the Stage I ceiling to express Stage II levels of MYC. Thus, the FUBPs enforce MYC setpoints and prevent excursions into Stage II. That KOKD cells become inviable after ~5 days suggests that beyond MYC regulation, the FUBPs may contribute to other essential cellular functions.

When treated with high levels of TNFα or PDGF, WT and *Fubp2* KD MEFs responded identically, their almost superimposable CDFs shifting rightward to enter stage II. With stimulation, *Fubp1*-KO MEFs moved to slightly higher expression than either WT or *Fubp2* KD across almost the full range of MYC expression. TNFα or PDGF stimulation of KOKD MEFs elicits the highest and most variable levels of MYC (Fig. 5d, e and Supplementary Fig. 6d, e) seen in these experiments. RNA in situ hybridization confirmed that this increased level of MYC protein in KOKD MEFs reflected a parallel increase of Myc mRNA (Fig. 5f). We conclude that together FUBP1 and FUBP2 help to confine the MYC-setpoint to lower mean values and to limit expression variability within stage I and reduce the range of MYC expression in Stage II.

**FUBP1 binds broadly to the *Myc* gene body and 3′-end in primary MEFs.** Although FUBP1 binds to FUSE in human cells, this element is imperfectly conserved in mice and differs in bases that would be predicted to attenuate binding of FUBP1 at the murine homologous location[38]. Considering the mega-base span of the *Myc* locus and as no assays have documented the profile of FUBP1 binding across the full locus in either mice or humans, we sought to survey FUBP1 binding in the *Myc* region using chromatin immunoprecipitation sequencing (ChIP-seq). Here we find that that FUBP1 binds broadly across the first exon and intron, and throughout the transcription termination region and the 3′-untranslated regions of the *Myc* gene in primary MEFs (Fig. 6a). FUBP1 also binds broadly across many other genes (Supplementary Fig. 7), which will be analyzed and published separately. Notably, the MYC coding region is bracketed by upstream and downstream FUBP1 peaks. Probing for FUBP1 binding at selected sites in *MYC* by ChIP-PCR yields a higher signal in *Fubp1* WT cells than in *Fubp1* KO cells (Supplementary Fig. 8a, b) (the residual binding likely represents cross-reaction of anti-FUBP1 with FUBP2 in ChIP (Supplementary Fig. 8c). When stimulated by PDGF or TNFα for 4 h, binding of FUBP1 to the *Myc* gene body increases compared with serum-starved cells (Fig. 6b), whereas the levels of FUBP1/2 are not notably affected (Supplementary Fig. 8d). Transcription through the *Myc* gene body would at least transiently expose the entire non-template strand to ssDNA binding proteins such as FUBP1. The frequency, life span, and function of such intragenic binding events remains to be evaluated.

A local maximum of FUBP1 binding ~2.5 kb upstream of the TSS was coincident with the strongest upstream predicted peak (mFUSE) of stress-induced duplex destabilization (SIDD)[59]; coincidence of FUBP1 binding and supercoiled-driven melting is a feature of the human FUSE region (Fig. 6a). Unlike humans, no predicted Z-DNA sites (a left-handed double helix with a 12 bp helical repeat) that would be stabilized by physiological levels of supercoiling were noted in the murine *Myc* gene. The

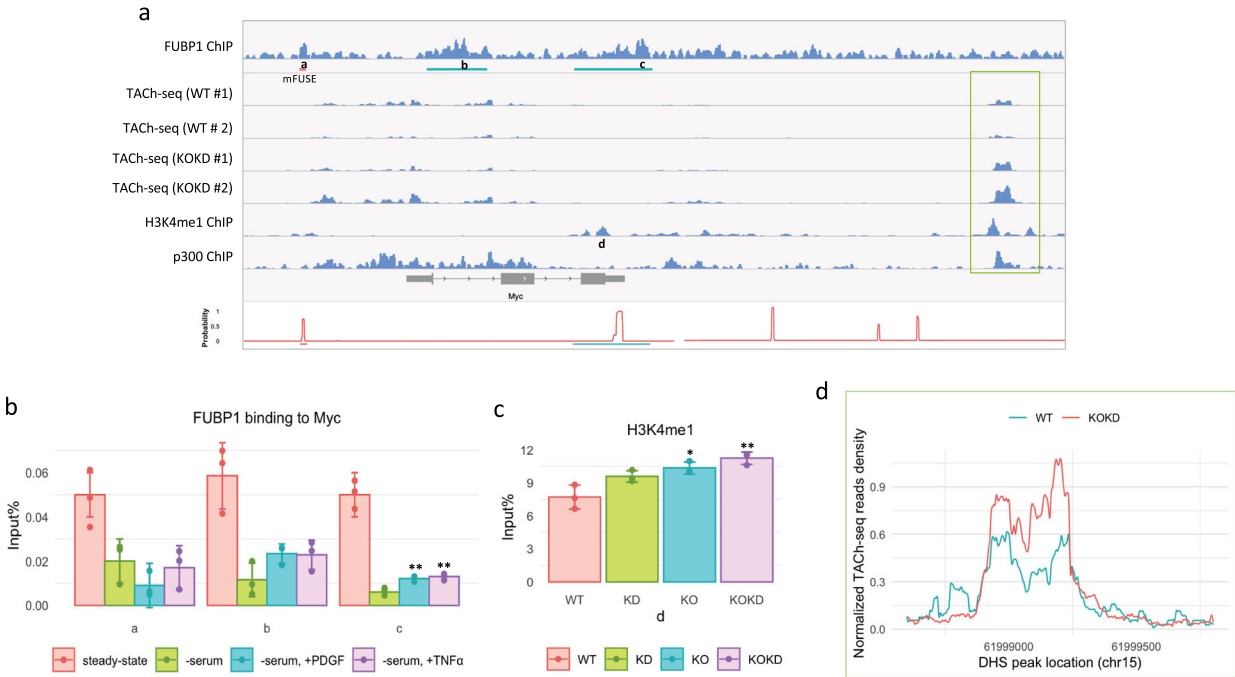

**Fig. 6 FUBP1 binds broadly to the Myc gene in parallel to the expression of MYC and loss of FUBPs causes changes in the chromatin at the Myc locus.**
**a** First panel: the FUBP1 ChIP-seq at *Myc* locus in MEFs. Peaks called by the SICER algorithm against input control are marked with green lines, FDR < 0.01. A mouse FUSE (mFUSE) with a high possibility of melting is marked by a pink line. The positions of the primers used in each panel of **b** and **c** were marked with the corresponding letters in this figure. The DNase hypersensitivity sites (DHS) downstream of *Myc* that were flanked by H3K4me1 peaks and overlapped with p300 peaks[84] are marked by a light green rectangle and analyzed in **d**. Second to the fifth lines: the DHS peaks identified by TACh-seq in WT and KOKD MEFs, biological duplicates. Sixth panel: ENCODE data (ENCFF592HMA) of H3K4me1 ChIP-seq in MEFs. Seventh panel: p300 ChIP-seq in MEFs from published literature. Bottom panel: the transition probabilities calculated by SIDD algorithm as functions of base pair at the *Myc* locus and regions susceptible to melting are shown. The sequences of 3.7 kb upstream of the transcription start site (TSS) to the 1.2 kb downstream of TTS was analyzed as a superhelical domain, whereas 1.2 to 10 kb downstream of TTS was analyzed as another superhelical domain. Superhelical density = −0.06. The positions of the primers used in each panel of **b** and **c** were marked with the corresponding letters. **b** FUBP1 ChIP-qPCR results of the *Myc* locus in steady-state and PDGF- or TNFα-stimulated MEFs are shown. The positions of the primers used in each panel are marked with the corresponding letters in **a**. Experiments were performed in triplicate. **p < 0.01, *T*-test assuming unequal variance, two-tailed, compared to the serum-starved sample. **c** The results of H3K4me1 ChIP-qPCR on the *Myc* gene in WT, KD, KO, and KOKD MEFs are shown. The position of the primers used in this figure is marked with the corresponding letters in Figure a. Experiments were performed in triplicate. *p < 0.05, **p < 0.01, *T*-test assuming unequal variance, two-tailed, compared to the WT. **d** The average of the normalized TACh-seq reads density of two biological replicates of WT and KOKD MEFs were plotted on the DHS peak downstream of *Myc* (the region marked by a light green rectangle in **a**). DESeq2 analysis showed the signal of this DHS peak in KOKD is significantly higher in KOKD than that in WT (FDR < 0.05), with an average fold increase of 1.6.

termination region 3′ of *Myc* also supports a strong SIDD site that would compete to absorb the torsional stress required for melting with mFUSE when both sequences reside in the same topological domain. During active transcription of *Myc*, mFUSE and the 3′-SIDD sites will not reside in the same topological domain as the transcribing RNAPII is a node that separates negative super-coiling upstream from positive supercoiling downstream of the enzyme. As the polymerase transits the 3′-SIDD sites, both mFUSE and the 3′-SIDD would become embraced within the same negatively supercoiled domain and compete for melting; most of the time, the 3′-SIDD will win this competition. FUBP1 binding to mFUSE and the 3′-downstream regions of MYC were compared in high- and low-MYC cells during steady-state growth (Supplementary Fig. 8e) and found to be parallel to MYC levels (Supplementary Fig. 8f). Previous studies of the human FUSE during MYC activation revealed that immediate FUBP3 recruit-ment that peaks 1 h post stimulation is followed at 2 h by FUBP1 that has largely declined by 4 h[45,60]. The collapse of mFUSE would evict FUBP1 until a subsequent round of activity regenerates enough torsional stress to repeat the mFUSE-FUBP1 regulatory cycle. Although a FUBP1 binding at mFUSE

declined 4 h after activation, FUBP1 remained at the coding and 3′-regions (Fig. 6b).

**Interrogating MYC for chromatin changes associated with high or low setpoints and/or FUBP action.** To assess the pos-sible involvement of processes that might relate to the epigenetic fixation of *MYC* setpoints, several features of chromatin were interrogated using ChIP-PCR and nuclease hypersensitivity ana-lysis. Multiple changes were noted, mainly in the gene body and 3′-downstream regions of the gene. First, histone H3K4me1, H3K4me3, and H3K27ac ChIP analyses were performed on *Fubp1* WT, KD, KO, and KOKD cells (Fig. 6c and Supplementary Fig. 9). PCR probes were selected to examine regions known to be enriched for these features (ENCODE[61,62] #ENCSR030BUT). We found that the 3′-region peak of H3K4me1 on the *Myc* gene in KO and KOKD cells is higher than that in WT cells. A dramatic increase in nuclease hypersensitivity was elicited in KOKD cells at a site 8 kb downstream of the TTS in a region that binds p300 (Fig. 6d). Together, these results indicate that the FUBPs directly or indirectly influence chromatin architecture at multiple loca-tions throughout the *MYC* locus.

 COMMUNICATIONS BIOLOGY | https://doi.org/10.1038/s42003-020-01264-x

## Discussion

Before the development of single-cell methods, gene expression in tissues and cell lines had been characterized with population-wide measurements—mean levels of protein and mRNA expression, and sometimes their half-lives. The biological interpretations of these measurements have typically assumed that the birth and death of these macromolecules were independent, single, Poisson-distributed stochastic events. With the advent of fluorescence- and -omics-based methods, it has become possible to observe and explain variability in gene expression between cells. Such variability has been considered to reflect either bona fide long-lived biological variation, as similar cells progressively canalize into distinct states and/or microstates[63], or, alternatively, the variability may arise due to intrinsic and extrinsic fluctuations within an otherwise homogeneous population[64]. Interrogating cell-to-cell variation of gene output by flow cytometry or quantitative fluorescence microscopy typically reveals a log-normal distribution of expression (a normal distribution of the logarithm of the measured parameter). Graphically, the log-normal distribution is a bell-shaped distribution with a long tail. The intercellular tallies of most proteins and RNAs exhibit such cell-to-cell variation spanning a range of one–two logs[65]. For many genes, such variation may be of little physiological or pathologic consequence; for some of these genes, such fluctuations may be simply averaged away over time, whereas for others the fluctuations may simply be tolerated with few lasting untoward consequences for imprecise expression control at the organismal level. With very short protein and mRNA half-lives, MYC would seem to be primed for fluctuating expression levels. Yet, MYC levels matter. Twofold differences in MYC levels have been associated with profound changes in cellular and organismal fates[13,19,22,23,29,30,66–68].

The plethora of regulatory mechanisms operating on MYC expression at all levels of macromolecular synthesis and degradation would argue that MYC levels are precisely regulated. By distributing regulation across multiple levels of macromolecule synthesis and degradation, expression variation is intrinsically reduced. In addition, there may be mechanisms that actively reduce expression noise. Indeed, it has been shown that FUBP1 can contribute to a feedforward/feedback system that regulates *Myc* transcription in real-time. FUBP1 has been proposed to be a key effector that can both up- or downregulate MYC and behave as a molecular cruise control that opposes *MYC* fluctuations[29,32,45,69]. The superposition of multiple layers on *Myc* regulation might argue that its expression has been selected to be more tightly controlled than most other genes. Yet, the evidence that MYC levels are indeed held to closer tolerances has been missing.

At first glance, MYC expression at steady state seems to be a typical log-normal-like distribution with protein levels spanning more than a decade. The bulk of cells reside in the narrow low-MYC peak where overall MYC-driven transcription amplification is likely to be modest. Other cells populate a long rightward tail where MYC is expected to be on duty. Yet, sorting and re-culturing the cells from either the low-MYC or the high-MYC sides of the distribution failed to repopulate the parental profile. Thus, the log-normal appearance of the total population belies a more precise setting of MYC expression and indicates that the observed distribution must be a superposition of cell-autonomous subpopulations with differing MYC setpoints. The maintenance of these setpoints through cell division and growth arrest indicates the operation of a somatic epigenetic mechanism. Although inhibitors of DNA methylation and histone deacetylation altered MYC levels, they acted equivalently on high- and low-MYC subpopulations and failed to erode the difference between the high and low cells. We surmise that the heritable fine-tuning of MYC may involve uncharacterized epigenetic mechanisms. The fact that HDAC inhibition, which generally augments

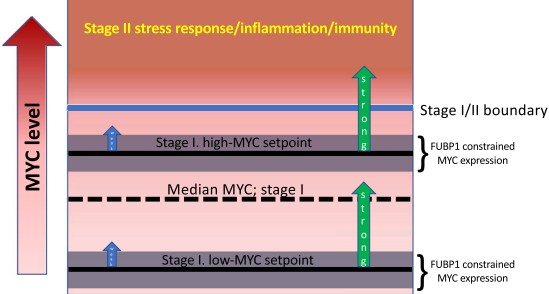

**Fig. 7 The basal levels of MYC determines the strength of the response to the stimulus.** In response to a weak stimulus (left), the cells in Stage I shift upwards while adhering to the upper limit. In response to a strong stimulus (right), the basal MYC levels determine the number of cells entering into Stage II. The cells with a high-MYC setpoint are more likely to enter Stage II. FUBP1 and FUBP2 play an essential role in limiting the expression of MYC in Stage I.

transcription, decreases MYC expression (here and as reported previously[70]) supports the existence of noncanonical epigenetic regulation of *Myc*.

We find that *Myc*-stimulation occurs in two stages in non-transformed B-cells and primary fibroblasts. In the first stage following treatment with low levels of growth factors (IL-4, FGF, PDGF, TNFα, etc.) and/or other mitogens (serum), unstressed cells shift mean MYC expression upward while respecting an upper boundary (Fig. 7). Such a regimen would seem to be well-suited for the physiological adjustments as low levels of MYC fine-tune steady-state expression.

When the cells are exposed to strong stimuli and express a large amount of MYC, the cells enter a second stage (Fig. 7). In Stage II, MYC expression rises acutely and becomes much more variable with a subfraction of cells displaying very high levels. Such variability has been shown to be functional in a variety of stress responses to environmental, inflammatory, or infectious challenges, where differences from one cell to the next are effectively hedged, allowing subsets of the cell population to find the optimal MYC levels. This variability is not entirely stochastic as the magnitude of the stage II response between subpopulations seems to be primed by their relative MYC setpoints in stage I. As under physiological conditions subpopulations of cells have different stage I setpoints, differing levels of stimulation may be required to drive the different subpopulations across the stage II boundary. Thus, the sensitivity to MYC-inducing signals may be tuned by adjusting the basal level of MYC. Such stable tuning of the Toll-like receptor response by MYC has been noted in mature B-cells[71,72]. The results here suggest that cells with higher basal MYC are primed to enter stage II and rapidly mount a robust stress response. As MYC amplifier-gain parallels increasing promoter output, these cells will preferentially increase their already higher levels of mediators of inflammation and immunity. High MYC stage I cells may thus serve as sentinels that alert and help to recruit their neighbors when they enter stage II (Fig. 7).

The proto-oncogene, *Myc*, becomes an oncogene through abnormal upregulation, not requiring changes in its protein sequence. Although coding mutations do occur, they primarily affect MYC stability and turnover, thereby increasing its levels rather than changing its fundamental action[73–76]. Much direct and indirect evidence suggests that MYC levels are important in proliferation and immortalization. Small changes in MYC levels prompt cell cycle-arrested cells to proliferate or to cross thresholds that trigger apoptosis[68]. When normal diploid human fibroblasts are immortalized by retroviral-transduced telomerase, these cells become senescent after knocking out a *c-Myc* allele so

small changes in MYC levels are associated with dramatically different cell fates[77]. Thus, how much MYC is oncogenic? The two-stage model presented here can help explain. In primary cells, MYC levels are in stage I and are all below the ceiling. When most cells break through the upper limit and enter the Stage II, it raises a hypothesis that this may be a sign of cell state change: cells may convert into another type of cell, or into a more active state, such as become active B-cells from naive B-cells, or become immortalized or carcinogenic. Quantitative and dynamic studies of MYC levels in different contexts are needed to test this hypothesis.

To limit MYC expression in Stage I, *Myc* is tightly regulated by many signaling pathways at the transcriptional, post-transcriptional and protein regulatory levels, and Myc mRNAs and proteins are relatively unstable. FUBP1/FIR/FUSE system has been proposed to be a molecular cruise control, limiting the upward and downward shift of MYC and possibly other FUBP1 target genes by providing both positive and negative feedback[48,49]. But this system has been less explored in mice. We have mapped the FUBP1-binding profile across the *Myc* gene. Here we report a candidate mFUSE ~2.5 kb upstream of the major *Myc* TSS; this element both melts in response to supercoiling and is a local maximum of FUBP1 binding. mFUSE is predicted to be topologically coupled with a SIDD site at the 3′-end of *Myc* when both elements reside in the same topological domain. Notably, upon loss of FUBP1 and especially when lacking both FUBP1 and 2, cells spontaneously enter Stage II. After stimulation with PDGF, binding at mFUSE is diminished 4 h after stimulation with PDGF. This finding suggests that the cruise-control action must be disengaged to accelerate and sustain high *MYC* output.

FUBP1 has been found to associate with a variety of genes that regulate cell growth or death besides MYC, including p21, Ccna1, and USP29, through binding to FUSE-like upstream regulatory sequences[78–80]. FUBP1 also binds to a broadly transcribed region of *Myc* and along its 3′-end in MEFs. As FUBP1 binds to a range of single-stranded nucleic acids across a hierarchy of affinities, this intragenic and 3′-region binding along with previous works[35,36,38,44–46,51,69,81,82] suggest that FUBP1 may regulate *Myc* transcription through conventional and unconventional incompletely characterized mechanisms that operate intragenically and at the 3′-end of *Myc*.

Among the three members of the FUBP family, FUBP2 shares ~65% sequence similarity with FUBP1 in human and mouse, and is co-expressed in almost every tissue with FUBP1, and thus their function may overlap. This study showed that a shared deficiency of both FUBP1 and FUBP2 dramatically increased both MYC levels and variability. The simultaneous loss of FUBP1 and FUBP2 may render the cells completely unable to recruit FIR (the remaining family member FUBP3 is unable to bind FIR), so that the cells lose both positive and negative feedback at the same time; therefore, the expression level will fluctuate greatly. Whether the overall level of expression rises or falls depends on the sum of the stimulatory or repressive signals of the cells at the time.

Looking ahead, the single-cell microscopy and characterization of MYC expression established in this study should instruct the further study of *Myc*, as well as of many other genes. Gene expression varies from cell to cell, and pathological deregulation of only a small subpopulation of cells may subsequently elicit disease. Treating gene levels in a cell group as a distribution rather than a single value may be the only way to understand the biology and pathology of these outliers (e.g., drug resistance and tumor relapse).

## Methods

**Cells and reagents**. Primary mouse embryonic fibroblasts were prepared as previously described[51]. Briefly, on gestation Day 13.5, pregnant female mice were killed and embryos were surgically separated, decapitated, and eviscerated. The bodies were minced, digested in trypsin for 10 min, and cells were plated in T-75 flasks. Then MEFs were cultured in Dulbecco's Modified Eagle Medium (Gibco # 10564) supplemented with 10% fetal bovine serum, 100 units/mL of penicillin, and 100 µg/mL of streptomycin (Gibco #15140) for within three passages. MEFs were 60~70% confluence before 4 h of stimulation and were 40~50% confluence before siRNA transfection. Primary resting naive mouse B-cells were isolated from splenocytes with anti-CD43 MicroBeads (Miltenyi Biotec) by negative selection. Antibodies to MYC (#ab32072) and FUBP1 (#ab184111) were from Abcam. LPS, mouse IL-4, and mouse FGF were from R&D Systems. Mouse PDGF was from Life Technology.

**Quantitative immunofluorescence and imaging acquisition**. Primary MEFs or primary naive B-cells were seeded in a fibronectin-coated 96-well glass-bottom plate (Thermo Scientific) and fixed with 2% paraformaldehyde. Cells were incubated with MYC primary antibody and a secondary antibody coupled to Alexa Fluor 488 (Thermo Fisher) and stained with Hoechst 33342 (Thermo Fisher). Cells were imaged on a Nikon Eclipse TiE inverted fluorescence microscope with a ×20 plan apo objective (NA 0.75) using an iXon Ultra-888 camera (Andor), and GFP (450–490/495/500–550 nm) and 4′,6-diamidino-2-phenylindole (DAPI) (325–375/400/435–485 nm) filter sets, to enable the collection of separate data from individual cells, as opposed to values averaged over populations of cells. At least 1000 cells were acquired per condition.

**Myc RNA in situ hybridization and imaging acquisition**. Primary MEFs were plated in 96-well plate, fixed with 2% paraformaldehyde, and stained with a fluorescent Myc RNA probe following the manufacturer's instructions (ACD). Cells were imaged on a Nikon Eclipse TiE inverted fluorescence microscope with a ×40 plan apo objective (NA 0.75) using an iXon Ultra-888 camera (Andor), and GFP (450–490/495/500–550 nm) and DAPI (325–375/400/435–485 nm) filter sets, to enable the collection of separate data from individual cells, as opposed to values averaged over populations of cells. At least 100 cells were acquired per condition.

**Flow cytometry analysis and cell sorting**. Primary MEFs from *Myc-Egfp*[+/+] mice were treated with 20 µM of MG-132 for 6 h, to inhibit proteasome degradation of MYC and therefore enhance the MYC-GFP signal, and then sorted for low- and high-GFP cells. The results were analyzed by Flowjo software.

**WST-1 proliferation assay**. The low-MYC and high-MYC cells after sorting were seeded in triplicate in a 100 µl volume. Ten microliters of WST-1 (Sigma) was added to the cells and absorbance was measured at 450 nm against 655 nm after 0.5 h.

**RNA sequencing**. After sorting, the low-MYC and high-MYC cells were seeded overnight. Considering the effects of MYC on global gene expression, external RNA spike-in control (ERCC, Thermo Fisher Scientific) was added to cells in proportion to cell count before total RNA extraction. The mRNA libraries were prepared using TruSeq Stranded mRNA (Illumina) and sequenced on a NextSeq sequencer.

**RNA interference**. WT and *Fubp1* KO MEFs were transfected with 100 nM of mouse *Fubp2* or non-target siRNA (Dharmacon) for 48 h using Dharmacon transfection reagents.

**ChIP-seq and ChIP-qPCR**. ChIP samples were prepared following regular ChIP protocol with minor modifications. Briefly, primary MEFs were cross-linked with 1% formaldehyde for 8 min (FUBP1 ChIP) or 2 min (H3K4me1, H3K4me3, and H3K27ac ChIP), and the cross-linking was terminated by adding glycine to a final concentration of 0.125 M. After collecting cells, the cell pellets were resuspended in RIPA buffer supplemented with protease inhibitors. Samples were sonicated to produce chromatin fragments of 300 bp on average and subjected to immuno-precipitation. Five micrograms of rabbit monoclonal anti-FUBP1 antibody (Abcam, ab181111), anti-H3K4me1 (Abcam, #ab8895), anti-H3K4me3 (Millipore, Sigma, #04-745), and anti-H3K27ac (ActiveMotif #39034) were used for the immunoprecipitation. The immunoprecipitated samples were subjected to library preparation and sequencing (ChIP-seq), or to qPCR quantification with SYBR Green. The ChIP-seq data sets for H3K4me1, H3K4me3, and H3K27ac in primary MEFs are from ENCODE data set ENCSR030BUT. The antibodies listed above for the histone marks ChIP-qPCR are the same as those used for the ENCODE data. The primers for FUBP1, H3K4me1, H3K4me3, and H3K27ac ChIP-qPCR on the *Myc* gene are listed in Supplementary Table 1.

**TACh sequencing**. The TACh sequencing (TACh-seq) was performed as previously described with modification[83]. Briefly, MEFs were lysed in hypertonic lysis buffer and digested with Cyanase (RiboSolutions) for 3 min at 37 °C. Reactions were terminated by the addition of EDTA (10 mM final) and SDS (0.75% final). After RNA and protein digestion, DNA was purified and the fragments of 100–600 bp were selected by gel electrophoresis. The sequencing libraries were prepared

using TruSeq Chip Sample Prep Kit (Illumina) and sequenced on a NextSeq sequencer.

**Immunoblotting.** Samples were lysed in radioimmunoprecipitation buffer for 30 min and 30 μg of total protein was used for immunoblotting.

*Quantitative immunofluorescence and Myc RNA in situ hybridization.* Image acquisition, automated segmentation, and intensity measurements were performed using Nikon Elements software version 4.30.02. For MYC protein immunostaining, the nuclear size indicated by Hoechst and the mean nuclear fluorescence intensity of MYC were measured in single cells, and the total amount of MYC protein was calculated as the product of mean intensity and the nuclear size. For Myc RNA in situ hybridization, the cell boundary of the bright-field image was determined by the Contour function provided by the Nikon Elements software and the cell size was determined. Total Myc mRNA intensity was calculated as the product of mean intensity and cell size. Data were analyzed and plotted using custom R software. The analysis codes are available from the corresponding author upon request.

*RNA-seq analysis.* RNA-seq data were processed using Partek Flow (Partek) and custom R code. After trimming bases on the 3′-end with quality scores below 20, reads were aligned to the mouse transcriptome (mm10) using TopHat. For each ERCC RNA, the number of reads mapping to that transcript was divided by the transcript length to yield Fragments Per Kilobase (FPK). From the top 50 expressed transcripts as measured by average FPK over all samples, a set of 10 highly correlated transcripts was used to represent the quantity of spike-in RNA in each sample. The normalization score for each sample was then calculated as the geometric mean of the expression (FPK) of those ten genes in that sample. Reads mapped to the transcriptome were quantified using the Partek E/M algorithm. For each transcript, the number of reads quantified to that transcript was divided by the transcript length to yield FPK, then divided by the normalization score for that sample to normalize to the number of cells. The normalized FPK of low-MYC and high-MYC cells were compared.

*ChIP-seq analysis.* After trimming bases on both ends with quality scores below 20, reads were aligned to the mouse genome (mm10) using Bowtie2. ChIP-seq peaks were called using SICER. The input DNA sequenced on the same lane was used as the control.

*ChIP-qPCR analysis.* The Cq of each ChIP sample was analyzed relative to the input, as this included normalization of background levels and the amount of chromatin for ChIP.

*TACh-seq analysis.* After trimming bases on both ends with quality scores below 20, reads were aligned to the mouse genome (mm10) using Bowtie2. The TACh-seq peaks were called using SICER. The difference in read density of the TACh-seq peaks between WT and KOKD MEFs was analyzed using DESeq2.

**Reporting summary.** Further information on research design is available in the Nature Research Reporting Summary linked to this article.

## Data availability

The RNA-seq, ChIP-seq, and TACh-seq sequencing data that support the findings of this study are available in GEO, accession number GSE143800. Source data for the main figures is provided in Supplementary Data 2. All other data that support the findings of this study are available from the corresponding author upon reasonable request.

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

## Acknowledgements

We thank Dr. Ferenc Livak in Flow Cytometry Core Facility (NCI, CCR) for assistance with the flow cytometry and sorting. We thank the sequencing facility (NCI, CCR) for assistance with next-generation sequencing. We thank the ENCODE Consortium and the Bing Ren lab (UCSD) generating the ENCSR030BUT data set. This work was supported by the Intramural Research Program of the National Cancer Institute, Center for Cancer Research.

## Author contributions

Y.Z. designed and performed experiments, analyzed results, performed computation, and wrote the manuscript. W.D. designed animal crosses and helped prepare experimental materials. C.J.B. wrote algorithms and analyzed DNA sequences. E.B. designed experiments, analyzed data, and edited the manuscript. D.L. designed experiments, analyzed results, and wrote the manuscript.

## Competing interests

The authors declare no competing interests.
