## [Peer Review File · Communications Biology]

Reviewers' comments:

Reviewer #1 (Remarks to the Author):

The manuscript by Zheng and colleagues is about single cells expression analysis of the MYC protein and transcript in primary B cells and in MEFs of WT mice and of FUBP1 KO mice. The authors perform single cells expression analysis using quantitative immunofluorescence microscopy. By stimulating resting cells with two different doses of growth factors the authors suggest that MYC induction is characterized by two stages: one in which MYC increase is due to an increase in the number of cells expressing low or intermediate MYC but not of the cells expressing high MYC; on the contrary in stage II there is the appearance of a cell population that express MYC levels higher than that observed in the top expressing cells in the unstimulated population. The same approach is used to investigate MYC expression in MEFs from FUBP1 KO mice treated or not with an siRNA to knock-down also FUBP2 and treated or not with either TNF α or PDGF. Next the authors use MEFs from a mouse model expressing a chimeric MYC-Egfp gene at the endogenous MYC locus. From a population of steady-state growing MEFs they sort cells expressing low and cells expressing high MYC and show that even after 7 day of culture both the low and high MYC cells still preserved their respective MYC expression levels. Differential gene expression analysis showed that up-regulated mRNAs in high-MYC cells were enriched for genes involved in the inflammatory response and the immune response compared to low-MYC cells. The authors also notice that MYC-high cells express more MYC upon restimulation relative to low-MYC cells but this process is not dependent on difference in DNA methylation and/or histone acetylation. Finally the authors use ChIP-seq to define the binding profile of FUSB1 on the MYC locus in mouse MEFs.

Generally what is missing in this manuscript is the definition of the aim of the study, the main biological question. This is particularly evident in the abstract that is basically a list of the results obtained. At the end of the Introduction the authors say that they want to... "examine whether there is a fixed set-point level of MYC and to observe its variation both in steady-state and stimulated primary cells, and to ask if the FUBPs contribute to this regulation". This point should be explained better. Otherwise it is difficult to judge if the experimental design of the manuscript is able to address its aim.

The first set of experiments use two primary cell models stimulated with two different doses of growth factors. Not a lot of information is given on the choice of the two doses. For IL4 the high dose is a 10 fold excess respect to the low dose; for FGF is a 4 fold excess. On which ground call the authors the stimulation with the low dose "physiological" and with high dose "pathological"? Next the authors use a statistical tool to define what they call "two stages" of MYC expression. The way the authors describe the increase in MYC expression following treatment with a high dose of growth factor suggest almost a phase transition in MYC expression. The question is: what happen if the authors test also all the other intermediate doses of growth factor: will they observe that each doses breach the maximum defined by the previous dose? Is it possible that there are indeed not two stages but multiple stages? Or the maximum is breached only after a certain amount of stimulus?

Next the authors use the same approach to investigate the changes in MYC expression in cells KO of FUBP1/2 treated or not with PDGF and/or TNF α . The observation that FUBP1 increases cell-to-cell variation in MYC levels is not new. The data reported in panel E of figure 3 of this manuscript is very similar to data reported in their previous publication in American Journal of Pathology (Figure 8, Panel A of: Far Upstream Element Binding Protein Plays a Crucial Role in Embryonic Development, Hematopoiesis, and Stabilizing Myc Expression Levels. Am J Pathol 2016,186: 701-715). That high doses of growth factors increase MYC expression even further is novel, but in a way expected and not so informative if not accompanied by mechanistic insights.

The only experiments that more directly address the question if there is a fixed set-point level of MYC in a cell population are those that make use of sorted MEFs expressing low or high EGFP-MYC levels. The observation that high-MYC cells are stable in time and tend to express more MYC upon re-stimulation is novel and interesting. But at the same time the authors fail to give a mechanistic explanation for this difference. They also do not provide a link with a physiological condition: are

the genes differentially expressed in MYC-high sorted MEFs also differentially expressed in vivo? Finally the experiment presented in Figure 5 is a characterization of the way FUBP1 binds to the MYC locus, but, without further analyses, does not help in giving an answer to the aim of the study.

Minor points:

- Page 6: Figure 2A is not about RNA intensity. Maybe the authors mean Supp.Fig 3a?
- Page 6: "Total Myc mRNA intensity was calculated as the product of mean intensity and cell size": how did the authors measure cell size?
- Page 7: "Upon stimulation with low concentrations of IL-4, the MYC distribution shifted rightward, without changing the peak values of the MYC expression." Do the authors refer to the peak value of a density plot similar to that shown in Figure 1? Maybe the authors could show the data in a box plot, similar to that shown in Figure 3.
- In panel 2B and 2C of Figure 2, what is the negative control?
- In the experiments presented in Figure 3 the authors use PDGF and TNF α as growth factors: why not using FGF as in the previous experiments in MEFs?
- Page 11: upon serum starvation of high-MYC cells, did the authors observe more apoptosis than in low-MYC cells?
- Pages 12 and 13: this last paragraph of the Results section should be moved to the Discussion section, as it is not the description of the result of an experiment.
- Page 18: the grammar of the sentence: "Gene expression varies from cell to cell, and pathologic deregulation of only a small subpopulation of cells, which may subsequently elicit disease" needs to be checked.
- Page 20: the authors should explain why they treat the MEFs from Myc-Egfp $^{+/+}$ mice with 20 μ M of MG-132 for 6 h before sorting.
- Figure 4F: the authors should explain the meaning of TNF α =1 and TNF α =5 as well as PDGF=1 and PDGF=5. This is not mentioned in the Figure legend. Are they nmolar concentrations? Are the same used in Figure 3? (Mention the concentrations in Figure 3 legend as well).

Reviewer #2 (Remarks to the Author):

The paper by Zheng et al., entitled: "FUBP1 and FUBP2 Enforce Distinct Epigenetic Setpoints for MYC Expression in Primary Single Cells" reports the results of a study concerning the level of Myc expression in single mouse embryo fibroblasts (MEFs).

The aim is to investigate the expression levels of Myc in a steady state MEFs population, in the absence of oncogene activation or in the presence of "low intensity" signals or "intense signaling". Two discrete populations expressing high and low Myc levels were sorted and characterised for proliferation rate, gene expression profiling and response to PDGF and TNF- α . Furthermore, the lack of ssDNA-binding proteins FUBP1 and FUBP2 leads to high Myc mRNA and protein levels. Finally, FUBP1 binding profile to the Myc gene by ChIPSeq in primary MEFs reveals multiples interaction sites.

This study provides novel and interesting information about the fluctuations of Myc level in MEFs at the single cell level, identifying two clearcut setpoints, defining the extent of PDGF- and TNF- α -dependent Myc modulation in the two populations, the increased expression of genes involved in the immune and inflammatory responses in the high-Myc population, the setpoints control by FUBP1,2 and finally, the multiple binding sites of FUBP1 to the murine Myc locus.

Despite the wealth of data, this paper is rather descriptive, with minimal mechanistic information. Little investigation is directed to the mechanism underlying the interesting bimodal Myc expression in the MEF population and the mechanistic relevance of the FUBP1/2 regulators to the Myc

setpoints. Also, no explanation/model are provided as to why the extent of PDGF- or TNF- α -dependent induction is higher in the high- than in the low-Myc expressing population.

More observation follow:

Methods

p.19: MEF were taken from mouse embryos at day 13.5. How long were they cultured afterwards? The culture conditions, cell density, timing and growth factors concentration in the culture media are relevant to Myc expression in MEFs. It has long been known that c-Myc protein levels vary inversely with the degree of cell confluency (J Clin Invest. 95:900-4,1995). Therefore, cell density and culturing time should be specified in the figure legends.

In Figure 1, panels a and c, is the percentage of Myc-positive cells in the population nearly 100%? To accurately pinpoint the low-Myc expressing population, the baseline level should be set by the aid of Myc $-/-$ MEFs.

In Figure 3, it is shown that the absence of FUBP1 and 2 causes increased Myc mRNA and protein levels with no partitioning in two discrete populations, in contrast with the loss of single factors that does not affect Myc setpoints. The authors conclude that FUBP1 and 2 reduce Myc expression in both stages. Which is the model envisaged to accommodate these data, considering that FUBP1 is considered a transcriptional activator of myc (Zhou et al, 2016)? How could be further tested?

In Figure 4, the two high and low Myc expressing populations are sorted and shown to maintain their respective Myc expression levels (4b and c). Both populations are characterised for their proliferation rate (4d), inflammatory and immune response gene expression (4e).

However, the authors did not test the two populations for the expression of FBP-1 and 2: if these DNA-binding factors are relevant to the Myc expression, they may level differently in the high versus low Myc population.

In panel F, high-Myc cells respond to a greater extent than low-Myc cells, following serum starvation and exposure to PDGF. Why high-Myc are primed to a greater response? The use of HDAC inhibitors was not helpful. Is FUBP1 level/binding to the Myc region different in the two populations, +/-PDGF?

Figure 5 reports the broad FUBP1 binding to the first exon/intron and to the 3' UTR region of the mouse Myc gene by ChIPSeq experiments. Do the extent/sites of FUBP1 binding change in the low versus high-Myc populations and in response to PDGF/TNF- α ?

Reviewer #3 (Remarks to the Author):

General comments

This manuscript questions the robustness and the range of MYC expression variation, at single cell level, in primary cells. The authors were able to separate two cell fractions of a MEF population, each of them characterized by their MYC expression level (low and high). They demonstrated that the low-MYC sub-population do not display MYC expression modulation upon stimulation. The high-MYC sub-population displayed an increase in MYC expression upon strong stimulation. Loss of combined FUBP1 and FUBP2 maximize this increase upon stimulation of the total cell population. On the contrary, the high-MYC sub-population showed a decrease in MYC expression upon exposure to DNA methylation agent and/or HDAC inhibitor, demonstrating an epigenetic regulation. The approaches are appropriate and the data are convincing. Still, the manuscript shows some weaknesses in some demonstrations and will gain impact if the authors were able to demonstrate the (direct or indirect) causative link between epigenetics modification and FUBP1/2 function(s).

Review

Major points :

-The authors have to justify more the use of MEF cells and naïve B cells for their study. In particular, they have to include references for data reporting what is known on those lineages for myc or fubp1/2, if those data exist. If those data do not exist, the authors have to present data on the level of expression of FUBP1 and FUBP1 on those primary cells at steady-state level, as well as upon stimulation and treatment (aza and TSA).

- The authors have to justify the choice of the low and high concentration of FGF, IL4, PDGF, TNF α . What was the rationale based on? For example, what are the known associated functions activated or inhibited at those concentrations?

- Explain also why low IL4 or low FGF is considered physiological and high FGF or IL4 pathological. Provide references, elaborate more on this point or reformulate.

- Very importantly, a clarification must be provided for the definition of stage I, stage II and MYC setpoint. Stage I and stage II are concepts that apply to a cell population since the stage I-stage II boundary is defined in the cumulative distributive fraction of the cell population. How do the authors integrate those concepts at a single cell level? Are the authors able to translate the stage I-stage II boundary into a clear threshold in Figure 3D (protein level) for instance? The authors have to rephrase the manuscript to redefine/clarify the stage I-II boundary (cell population) or threshold (single cell). If there is no difference, use the same word. Clarify the definition of MYC-setpoint compared to the stage I-II boundary.

- Associated to Figure 3, it is stated "... to reduce the range of MYC expression in Stage II ». This statement is not appropriately quantified or demonstrated. In fig 3D, for instance, the range of MYC expression does not seem a lot different between WT and KOKD conditions. To me, the range of MYC expression looks the same, it is the number of cells expressing high-MYC that seem to be different. Could the author make the manuscript more clear on this point ?

- It would be interesting to show the proliferation status of KD, KO and KOKD cells compared to WT cells, upon treatment and non treated cells in order to draw correlation between the level MYC, level of FUBP1/2 and the proliferation rate. At least two publications (Rabenhorst, Cell Report 2015 and Debaize, Nucleic Acid Research, 2018) demonstrate that FUBP1 sustains proliferation (in mouse or human).

- Data from Rabenhorst, Cell Report 2015 show that FUBP1 shRNA decreases the level of Myc mRNA. How do the authors also articulate their results with the report showing that FUBP1 alone is not sufficient to activate MYC expression, but it is required for its maximal activation (He L, Liu J, Collins I et al (2000) Loss of FBP function arrests cellular proliferation and extinguishes c-myc expression. EMBO J 19:1034–1044) ? Could the authors comment/discuss more on those apparently contradictory data with their Fig 3A and E ?

- To estimate the specificity of the binding of FUBP1 on the Myc locus, it would be informative to show chip-seq data on other genes known to be regulated by FUBP1/2 and genes that are not. Indeed, what is striking here is the broad FUBP1 signal in the myc locus. Is FUBP1 bound everywhere like this in the genome? Can the authors provide some whole-genome analysis of their Fubp1-Chip-Seq and comment on it? The authors could also perform Chip-Seq (or Chip-qPCR) on the myc locus in MEF KD, KO and KOKD cells to demonstrate the implication of FUBP1/2 binding at those regions on the genome and MYC expression variability. The same after TNF or PDGF stimulation.

- The causative link between FUBP1/2 and epigenetically-controlled Myc-setpoint is not clearly demonstrated (see the title). For example, are the levels of H3K27ac, H3K4me1, H3K4me3 modified at this locus in MEF KD, KO and KOKD cells? or, is the level of MYC sensitive to aza or

TSA in MEF KD, KO and KOKD?

-The text and data are minimalists. Important information is lacking. Among them:

(1) The numbers of replicates must be indicated for each figure. Please clearly state whether or not all the graphs with 'fraction of cells' represent one experiment or are the sum of many (and indicated the number).

(2) The following data are required and must be displayed in supplemental figures: western blots allowing estimating the efficacy of extinction of Fubp1 and Fubp2 in KD and KO.

(3) Show the density curves corresponding to Fig2B and C, in supplemental data (those representation are more usual), corresponding to all the conditions (negative, untreated, low and high). Doing so, the 'peak value' will be clear.

- (4) Statistics should be done and indicated when appropriate on each figure.

(5) Provide sufficient information (or appropriate references) on ChIP-seq in the material and method so that the experiment can be repeated by other labs. Provide also the external link to the deposit of whole ChIP-Seq data.

(6) The characterization of MEF myc-egfp cells or mice needs to be provided, including the homozygotic status.

Minor points:

- In the abstract: the causative link between FUBPs and epigenetic setpoints has not been fully demonstrated. Moreover, in absence of stimulation, it seems to be exaggerated to write "Cells lacking ssDNA-binding proteins FUBP1 and FUBP2 overpopulate stage II ". Thus these two sentences should be edited. « Cells lacking ssDNA-binding proteins FUBP1 and FUBP2 overpopulate stage II, even when unstimulated. Thus, the FUBPs help to enforce constraints on the epigenetic setpoints that restrict MYC expression to stage I. ».

- B naive cells were used only in Figure 2, and not in the rest of the manuscript. This result is not of major interest for this manuscript. I suggest transferring those B-cell data in the supplemental part of the manuscript. Data on RNA level of MYC in naïve B cells are also lacking.

- justify the change of compound used to stimulate the cells since FGF is no longer used in MEF cells.

- In the legend of Fig3, the concentration of PDGF and TNF must be indicated.

- in the discussion, the authors can comment on plausible direct MYC-target genes among the list on modulated genes (supplemental Table 1).

- The authors may also discuss on another role of FUBP1/2 that could be implicated in the control of the MYC-setpoint, by participating in different promoter-enhancer interaction. (and where some enhancer can be accessible or not by DNA methylation or Histone modification).

The manuscript by Zheng and colleagues is about single cells expression analysis of the MYC protein and transcript in primary B cells and in MEFs of WT mice and of FUBP1 KO mice. The authors perform single cells expression analysis using quantitative immunofluorescence microscopy. By stimulating resting cells with two different doses of growth factors the authors suggest that MYC induction is characterized by two stages: one in which MYC increase is due to an increase in the number of cells expressing low or intermediate MYC but not of the cells expressing high MYC; on the contrary in stage II there is the appearance of a cell population that express MYC levels higher than that observed in the top expressing cells in the unstimulated population. The same approach is used to investigate MYC expression in MEFs from FUBP1 KO mice treated or not with an siRNA to knock-down also FUBP2 and treated or not with either TNF α or PDGF. Next the authors use MEFs from a mouse model expressing a chimeric MYC-Egfp gene at the endogenous MYC locus. From a population of steady-state growing MEFs they sort cells expressing low and cells expressing high MYC and show that even after 7 day of culture both the low and high MYC cells still preserved their respective MYC expression levels. Differential gene expression analysis showed that up-regulated mRNAs in high-MYC cells were enriched for genes involved in the inflammatory response and the immune response compared to low-MYC cells. The authors also notice that MYC-high cells express more MYC upon restimulation relative to low-MYC cells but this process is not dependent on difference in DNA methylation and/or histone acetylation. Finally the authors use ChIP-seq to define the binding profile of FUSB1 on the MYC locus in mouse MEFs.

Generally what is missing in this manuscript is the definition of the aim of the study, the main biological question. This is particularly evident in the abstract that is basically a list of the results obtained. At the end of the Introduction the authors say that they want to... "examine whether there is a fixed set-point level of MYC and to observe its variation both in steady-state and stimulated primary cells, and to ask if the FUBPs contribute to this regulation". This point should be explained better. Otherwise it is difficult to judge if the experimental design of the manuscript is able to address its aim.

The first set of experiments use two primary cell models stimulated with two different doses of growth factors. Not a lot of information is given on the choice of the two doses. For IL4 the high dose is a 10 fold excess respect to the low dose; for FGF is a 4 fold excess. On which ground call the authors the stimulation with the low dose "physiological" and with high dose "pathological"? Next the authors use a statistical tool to define what they call "two stages" of MYC expression. The way the authors describe the increase in MYC expression following treatment with a high dose of growth factor suggest almost a phase transition in MYC expression. The question is: what happen if the authors test also all the other intermediate doses of growth factor: will they observe that each doses breach the maximum defined by the previous dose? Is it possible that there are indeed not two stages but multiple stages? Or the maximum is breached only after a certain amount of stimulus?

Next the authors use the same approach to investigate the changes in MYC expression in cells KO of FUBP1/2 treated or not with PDGF and/or TNF α . The observation that FUBP1 increases cell-to-cell variation in MYC levels is not new. The data reported in panel E of figure 3 of this manuscript is very similar to data reported in their previous publication in American Journal of Pathology (Figure 8, Panel A of: Far Upstream Element Binding Protein Plays a Crucial Role in Embryonic Development, Hematopoiesis, and Stabilizing Myc Expression Levels. *Am J Pathol* 2016,186: 701-715). That high

doses of growth factors increase MYC expression even further is novel, but in a way expected and not so informative if not accompanied by mechanistic insights.

The only experiments that more directly address the question if there is a fixed set-point level of MYC in a cell population are those that make use of sorted MEFs expressing low or high EGFP-MYC levels. The observation that high-MYC cells are stable in time and tend to express more MYC upon re-stimulation is novel and interesting. But at the same time the authors fail to give a mechanistic explanation for this difference. They also do not provide a link with a physiological condition: are the genes differentially expressed in MYC-high sorted MEFs also differentially expressed *in vivo*?

Finally the experiment presented in Figure 5 is a characterization of the way FUBP1 binds to the MYC locus, but, without further analyses, does not help in giving an answer to the aim of the study.

Minor points:

- Page 6: Figure 2A is not about RNA intensity. Maybe the authors mean Supp.Fig 3a?
- Page 6: “Total Myc mRNA intensity was calculated as the product of mean intensity and cell size”: how did the authors measure cell size?
- Page 7: “Upon stimulation with low concentrations of IL-4, the MYC distribution shifted rightward, without changing the peak values of the MYC expression.” Do the authors refer to the peak value of a density plot similar to that shown in Figure 1? Maybe the authors could show the data in a box plot, similar to that shown in Figure 3.
- In panel 2B and 2C of Figure 2, what is the negative control?
- In the experiments presented in Figure 3 the authors use PDGF and TNF α as growth factors: why not using FGF as in the previous experiments in MEFs?
- Page 11: upon serum starvation of high-MYC cells, did the authors observe more apoptosis than in low-MYC cells?
- Pages 12 and 13: this last paragraph of the Results section should be moved to the Discussion section, as it is not the description of the result of an experiment.
- Page 18: the grammar of the sentence: “Gene expression varies from cell to cell, and pathologic deregulation of only a small subpopulation of cells, which may subsequently elicit disease” needs to be checked.
- Page 20: the authors should explain why they treat the MEFs from Myc-Egfp $^{+}/+$ mice with 20 μ M of MG-132 for 6 h before sorting.
- Figure 4F: the authors should explain the meaning of TNF α =1 and TNF α =5 as well as PDGF=1 and PDGF=5. This is not mentioned in the Figure legend. Are they nmolar concentrations? Are the same used in Figure 3? (Mention the concentrations in Figure 3 legend as well).

The paper by Zheng et al., entitled: "**FUBP1 and FUBP2 Enforce Distinct Epigenetic Setpoints for MYC Expression in Primary Single Cells**" reports the results of a study concerning the level of Myc expression in single mouse embryo fibroblasts (MEFs).

The aim is to investigate the expression levels of Myc in a steady state MEFs population, in the absence of oncogene activation or in the presence of "low intensity" signals or "intense signaling". Two discrete populations expressing high and low Myc levels were sorted and characterised for proliferation rate, gene expression profiling and response to PDGF and TNF- α . Furthermore, the lack of ssDNA-binding proteins FUBP1 and FUBP2 leads to high Myc mRNA and protein levels. Finally, FUBP1 binding profile to the Myc gene by ChIPSeq in primary MEFs reveals multiples interaction sites.

This study provides novel and interesting information about the fluctuations of Myc level in MEFs at the single cell level, identifying two clearcut setpoints, defining the extent of PDGF- and TNF- α -dependent Myc modulation in the two populations, the increased expression of genes involved in the immune and inflammatory responses in the high-Myc population, the setpoints control by FUBP1,2 and finally, the multiple binding sites of FUBP1 to the murine Myc locus.

Despite the wealth of data, this paper is rather descriptive, with minimal mechanistic information. Little investigation is directed to the mechanism underlying the interesting bimodal Myc expression in the MEF population and the mechanistic relevance of the FUBP1/2 regulators to the Myc setpoints. Also, no explanation/model are provided as to why the extent of PDGF- or TNF- α -dependent induction is higher in the high- than in the low-Myc expressing population.

More observation follow:

Methods

p.19: MEF were taken from mouse embryos at day 13.5. How long were they cultured afterwards? The culture conditions, cell density, timing and growth factors concentration in the culture media are relevant to Myc expression in MEFs. It has long been known that c-Myc protein levels vary inversely with the degree of cell confluency (J Clin Invest. 95:900-4,1995). Therefore, cell density and culturing time should be specified in the figure legends.

In Figure 1, panels a and c, is the percentage of Myc-positive cells in the population nearly 100%? To accurately pinpoint the low-Myc expressing population, the baseline level should be set by the aid of Myc -/- MEFs.

In Figure 3, it is shown that the absence of FUBP1 and 2 causes increased Myc mRNA and protein levels with no partitioning in two discrete populations, in contrast with the loss of single factors that does not affect Myc setpoints. The authors conclude that FUBP1 and 2 reduce Myc expression in both stages. Which is the model envisaged to accomodate these data, considering that FUBP1 is considered a transcriptional activator of myc (Zhou et al, 2016)? How could be further tested?

In Figure 4, the two high and low Myc expressing populations are sorted and shown to maintain their respective Myc expression levels (4b and c). Both populations are characterised for their proliferation rate (4d), inflammatory and immune response gene expression (4e).

However, the authors did not test the two populations for the expression of FBP-1 and 2: if these DNA-binding factors are relevant to the Myc expression, they may level differently in the high versus low Myc population.

In panel F, high-Myc cells respond to a greater extent than low-Myc cells, following serum starvation and exposure to PDGF. Why high-Myc are primed to a greater response? The use of HDAC inhibitors was not helpful. Is FUBP1 level/binding to the Myc region different in the two populations, +/-PDGF?

Figure 5 reports the broad FUBP1 binding to the first exon/intron and to the 3' UTR region of the mouse Myc gene by ChIPSeq experiments. Do the extent/sites of FUBP1 binding change in the low versus high-Myc populations and in response to PDGF/TNF- α ?

Responses to the reviewers' critiques:

Response to Reviewer 1

Reviewer #1 (Remarks to the Author):

The manuscript by Zheng and colleagues is about single cells expression analysis of the MYC protein and transcript in primary B cells and in MEFs of WT mice and of FUBP1 KO mice. The authors perform single cells expression analysis using quantitative immunofluorescence microscopy. By stimulating resting cells with two different doses of growth factors the authors suggest that MYC induction is characterized by two stages: one in which MYC increase is due to an increase in the number of cells expressing low or intermediate MYC but not of the cells expressing high MYC; on the contrary in stage II there is the appearance of a cell population that express MYC levels higher than that observed in the top expressing cells in the unstimulated population. The same approach is used to investigate MYC expression in MEFs from FUBP1 KO mice treated or not with an siRNA to knock-down also FUBP2 and treated or not with either TNF α or PDGF. Next the authors use MEFs from a mouse model expressing a chimeric MYC-Egfp gene at the endogenous MYC locus. From a population of steady-state growing MEFs they sort cells expressing low and cells expressing high MYC and show that even after 7 day of culture both the low and high MYC cells still preserved their respective MYC expression levels. Differential gene expression analysis showed that up-regulated mRNAs in high-MYC cells were enriched for genes involved in the inflammatory response and the immune response compared to low-MYC cells. The authors also notice that MYC-high cells express more MYC upon restimulation relative to low-MYC cells but this process is not dependent on difference in DNA methylation and/or histone acetylation. Finally the authors use ChIP-seq to define the binding profile of FUSB1 on the MYC locus in mouse MEFs.

Generally what is missing in this manuscript is the definition of the aim of the study, the main biological question. This is particularly evident in the abstract that is basically a list of the results obtained. At the end of the Introduction the authors say that they want to... "examine whether there is a fixed set-point level of MYC and to observe its variation both in steady-state and stimulated primary cells, and to ask if the FUBPs contribute to this regulation". This point should be explained better. Otherwise it is difficult to judge if the experimental design of the manuscript is able to address its aim.

In both the abstract and in the introduction, we have attempted to better enumerate the aims of our study and to better define the questions that we hope to address.

The first set of experiments use two primary cell models stimulated with two different doses of growth factors. Not a lot of information is given on the choice of the two doses. For IL4 the high dose is a 10 fold excess respect to the low dose; for FGF is a 4 fold excess. On which ground call the authors the stimulation with the low dose “physiological” and with high dose “pathological”? Next the authors use a statistical tool to define what they call “two stages” of MYC expression. The way the authors describe the increase in MYC expression following treatment with a high dose of growth factor suggest almost a phase transition in MYC expression. The question is: what happen if the authors test also all the other intermediate doses of growth factor: will they observe that each doses breach the maximum defined by the previous dose? Is it possible that there are indeed not two stages but multiple stages? Or the maximum is breached only after a certain amount of stimulus?

In the preparatory experiments leading to the work presented in this manuscript, we performed cytokine and growth-factor curves to define the concentrations that yield the maximal-MYC induction and maximal growth using low passage primary MEFs. At the concentrations employed apoptosis was not observed. The factors used, IL-4, PDGF, FGF, and TNF α , are all induced by stress, inflammation, infection, and/or neoplasia—all pathologic processes. We do not use the term “pathologic” to assert that there is pathologic over-production of these factors, but rather that the levels that sponsor maximal MYC expression are beyond the range associated with unstressed tissue homeostasis. We now *operationally* define physiological as a weak/homeostatic response and “stressed/pathological” as an intense/maximal response to a major perturbation. The insight of reviewer 3 is appreciated as a review of the literature indeed complicates a strict definition of “normal” versus “pathologic”. Normal and pathologic are defined using a bewildering array of different experimental systems, cell-lines, diseases, and assays for cytokine and growth factor levels and function. However common to all of them (and we spent a lot of time scouring the literature) is that the statistical separation between normal versus pathologic typically occurs over a growth-factor/cytokine concentration range of two to four-fold—the same sort of range that we empirically use for IL-4, TNF α , PDGF and FGF. (It should be noted that in some human tumors, FGF levels may become elevated more than 10x). Notably, the same concentrations of PDGF and TNF α concentrations that we employed separate normal and pathologic levels in at least one study (Andrade Junior *et al*, 2008; Koizumi *et al*, 2015).

The question of whether there might more than two stages, and indeed multiple stages is interesting, but problematic to establish. While stage I includes the majority of cells which weakly activate MYC, stage II is populated by a smaller population that induces MYC to much higher levels. Therefore, further subdivision of stages I and II would require many additional experiments that analyze a much greater number of cells in order to achieve the statistical resolving power necessary to sustain such substratification. We believe that the use of two-stages is conservative, but still establishes the concepts.

Next the authors use the same approach to investigate the changes in MYC expression in cells KO of FUBP1/2 treated or not with PDGF and/or TNF α . The observation that FUBP1 increases cell-to-cell variation in MYC levels is not new. The data reported in panel E of figure 3 of this manuscript is very similar to data reported in their previous publication in American Journal of Pathology (Figure 8, Panel A of: Far Upstream Element Binding Protein Plays a Crucial Role in Embryonic Development, Hematopoiesis, and Stabilizing Myc Expression Levels. Am J Pathol 2016,186: 701-715). That high doses of growth factors increase MYC expression even further is novel, but in a way expected and not so informative if not accompanied by mechanistic insights.

We agree that the increased variation in MYC levels elicited by growth factors and cytokines shown in previous Figure 2 is not surprising. But this result must be presented as a prelude to **Figure 3** which shows unexpectedly that the increased variation is not due to greater MYC fluctuation within an expanded *single* population, but in fact results from distinct sub-populations with different mean setpoints for MYC. **Figure 2** thus contrasts with **Figure 3**. In the revised manuscript we begin to address the epigenetic basis of this phenomenon and have identified some chromatin changes associated with setpoint fixation. We believe that this is important and interesting finding merits presentation to a broad community of biologists who may find other important examples of setpoint fixation and/or provide further insight on how it occurs.

The only experiments that more directly address the question if there is a fixed set-point level of MYC in a cell population are those that make use of sorted MEFs expressing low or high EGFP-MYC levels. The observation that high-MYC cells are stable in time and tend to express more MYC upon re-stimulation is novel and interesting. But at the same time the authors fail to give a mechanistic explanation for this difference. They also do not provide a link with a physiological condition: are the genes differentially expressed in MYC-high sorted MEFs also differentially expressed in vivo?

Finally the experiment presented in Figure 5 is a characterization of the way FUBP1 binds to the MYC locus, but, without further analyses, does not help in giving an answer to the aim of the study.

In the revised manuscript we examine further the expression differences between the high- and low-MYC cells. The dramatic enrichment of inflammation- and immunity-related genes expressed in the high-MYC cells coupled with their greater induction of MYC suggests that these cells are primed to mount a vigorous response to pathologic challenge. We suggest that higher MYC levels in the stage II subpopulation allow these “sentinel cells” that rapidly amplify the expression of cytokines and chemokines to alert their neighbors of an incipient threat. This is now discussed on **Pages 11 and 19** of the revised manuscript. We hope that the editors and reviewers agree that the validation of this concept in different tissues in vivo is beyond the scope of the current study.

Minor points:

- Page 6: Figure 2A is not about RNA intensity. Maybe the authors mean Supp.Fig 3a?

Thank you for pointing this out.

- Page 6: “Total Myc mRNA intensity was calculated as the product of mean intensity and cell size”: how did the authors measure cell size?

We have added this information to the Materials and Methods on **Page 26**. “For Myc RNA in situ hybridization, the cell boundary of the bright field image was determined by the “Contour” function provided by the Nikon Elements software, and the cell size was determined. Total *Myc* mRNA intensity was calculated as the product of mean intensity and cell size.”

- Page 7: “Upon stimulation with low concentrations of IL-4, the MYC distribution shifted rightward, without changing the peak values of the MYC expression.” Do the authors refer to the peak value of a density plot similar to that shown in Figure 1? Maybe the authors could show the data in a box plot, similar to that shown in Figure 3.

We have addressed this point by showing the density plots of the IL-4 treated B-cells and the FGF treated MEFs in **Supplemental Figure 2** and report these results on **Page 8** of the revised text.

- In panel 2B and 2C of Figure 2, what is the negative control?

For those experiments, non-immune IgG isotype control was compared with anti-MYC as the primary antibody. *Myc*^{-/-} cells are not viable and so could not be used as a negative control. We’ve added the negative control information to the legend of both **Figure 2** and **Supplemental Figure 2**.

- In the experiments presented in Figure 3 the authors use PDGF and TNF α as growth factors: why not using FGF as in the previous experiments in MEFs?

FGF was used and similar results were obtained. We’ve added the CDF curve and box plot in **Supplemental Figures 5B** and **5E**, to compare the steady-state and FGF-treated samples.

- Page 11: upon serum starvation of high-MYC cells, did the authors observe more apoptosis than in low-MYC cells? No, we did not.

- Pages 12 and 13: this last paragraph of the Results section should be moved to the Discussion section, as it is not the description of the result of an experiment.

Yes, this paragraph may not be a description of the result; however, it gives context for the result of mFUSE identification.

- Page 18: the grammar of the sentence: “Gene expression varies from cell to cell, and pathologic deregulation of only a small subpopulation of cells, which may subsequently elicit disease” needs to be checked.

Thank you. It has been corrected.

- Page 20: the authors should explain why they treat the MEFs from Myc-Egfp^{+/+} mice with 20 μ M of MG-132 for 6 h before sorting.

We’ve added the explanation to **Page 24**, “Primary MEFs from Myc-Egfp^{+/+} mice were treated with 20 μ M of MG-132 for 6 h to inhibit proteasome degradation of MYC and therefore enhance the MYC-GFP signal, and then sorted for low- and high-GFP cells.”

- Figure 4F: the authors should explain the meaning of TNF α =1 and TNF α =5 as well as PDGF=1 and PDGF=5. This is not mentioned in the Figure legend. Are they nmolar concentrations? Are the same used in Figure 3? (Mention the concentrations in Figure 3 legend as well).

Thank you. It is ng/ml. We’ve added the unit of stimuli to the figure legends of **Figures 2, 4, 5** and **Supplemental Figures 2 and 5**.

Andrade Junior DR, Santos SA, Castro I, Andrade DR (2008) Correlation between serum tumor necrosis factor alpha levels and clinical severity of tuberculosis. *Braz J Infect Dis* 12: 226-233

Koizumi T, Komiyama N, Nishimura S (2015) In-Vivo Higher Plasma Levels of Platelet-Derived Growth Factor and Matrix Metalloproteinase-9 in Coronary Artery at the Very Onset of Myocardial Infarction with ST-Segment Elevation. *Ann Vasc Dis* 8: 297-301

Response to Reviewer 2

Reviewer #2 (Remarks to the Author):

The paper by Zheng et al., entitled: "FUBP1 and FUBP2 Enforce Distinct Epigenetic Setpoints for MYC Expression in Primary Single Cells" reports the results of a study concerning the level of Myc expression in single mouse embryo fibroblasts (MEFs).

The aim is to investigate the expression levels of Myc in a steady state MEFs population, in the absence of oncogene activation or in the presence of “low intensity” signals or “intense signaling”. Two discrete populations expressing high and low Myc levels were sorted and characterized for proliferation rate, gene expression profiling and response to PDGF and TNF- α . Furthermore, the lack of ssDNA-binding proteins FUBP1 and FUBP2 leads to high Myc mRNA and protein levels. Finally, FUBP1 binding profile to the Myc gene by ChIPSeq in primary MEFs reveals multiples interaction sites.

This study provides novel and interesting information about the fluctuations of Myc level in MEFs at the single cell level, identifying two clearcut setpoints, defining the extent of PDGF- and TNF- α -dependent Myc modulation in the two populations, the increased expression of genes involved in the immune and inflammatory responses in the high-Myc population, the setpoints control by FUBP1,2 and finally, the multiple binding sites of FUBP1 to the murine Myc locus.

Despite the wealth of data, this paper is rather descriptive, with minimal mechanistic information. Little investigation is directed to the mechanism underlying the interesting bimodal Myc expression in the MEF population and the mechanistic relevance of the FUBP1/2 regulators to the Myc setpoints. Also, no explanation/model are provided as to why the extent of PDGF- or TNF- α -dependent induction is higher in the high- than in the low-Myc expressing population.

We have made numerous changes to the text and figures to address the reviewer’s critique. We have identified chromatin features that distinguish high- versus low-MYC and FUBP1 wild-type versus KOKD MEFs. This has been added to **Figure 6** and is discussed on **Page 15**. We now consider that increased MYC-driven transcription amplification, especially of inflammation/immunity genes in high-MYC cells may enable these cells to act as “sentinels” for stress and pathologic processes. (**Page 11**).

More observation follow:

Methods

p.19: MEF were taken from mouse embryos at day 13.5. How long were they cultured afterwards? The culture conditions, cell density, timing and growth factors concentration in the culture media are relevant to Myc expression in MEFs. It has long been known that c-Myc protein levels vary inversely with the degree of cell confluency (J Clin Invest. 95:900-4,1995). Therefore, cell density and culturing time should be specified in the figure legends.

The MEF preparation and culture conditions have now been described in the Materials and Methods on **Page 22**.

In Figure 1, panels a and c, is the percentage of Myc-positive cells in the population nearly 100%? To accurately pinpoint the low-Myc expressing population, the baseline level should be set by the aid of *Myc*^{-/-} MEFs.

While we appreciate this point, *Myc*^{-/-} MEFs are inviable and so cannot be used as a negative control. Non-immune IgG isotype control was compared with anti-MYC as the primary antibody. We've added the negative control information to the legend of both **Figure 2** and **Supplemental Figure 2**.

In Figure 3, it is shown that the absence of FUBP1 and 2 causes increased Myc mRNA and protein levels with no partitioning in two discrete populations, in contrast with the loss of single factors that does not affect Myc setpoints. The authors conclude that FUBP1 and 2 reduce Myc expression in both stages.

We do not claim that the FUBPs affect MYC setpoints, rather we find that the FUBPs enforce setpoints fixed by other mechanisms. In the new manuscript (**Pages 12-14**) we deal with this issue.

Which is the model envisaged to accommodate these data, considering that FUBP1 is considered a transcriptional activator of *myc* (Zhou et al, 2016)? How could be further tested?

In previous studies, we have provided evidence in support of the hypothesis that FUBP1 is a molecular cruise-control that dampens both upward and downward fluctuations in MYC levels. Just as a cruise-control must actuate both the accelerator and the brake, depending on circumstances, so FUBP1 can either up- or downregulate transcription. This may also account for FUBP1 being both prooncogenic and a tumor suppressor. The cruise-control concept is discussed on **Page 17** of the revised manuscript.

In Figure 4, the two high and low Myc expressing populations are sorted and shown to maintain their respective Myc expression levels (4b and c). Both populations are characterised for their proliferation rate (4d), inflammatory and immune response gene expression (4e).

However, the authors did not test the two populations for the expression of FBP-1 and 2: if these DNA-binding factors are relevant to the Myc expression, they may level differently in the high versus low Myc population.

The mRNA-seq on the low- and high-MYC cells shows there is a significant increase in gene expression in high-MYC cells compared to low-MYC cells and almost no downregulated genes. In parallel with the increase of MYC, *Fubp1* and *Fubp2/Khsrp* mRNA in high-MYC cells increased about 2 times than those in low-MYC cells (**Figure for the Reviewers 1**).

In panel F, high-Myc cells respond to a greater extent than low-Myc cells, following serum starvation and exposure to PDGF. Why high-Myc are primed to a greater response?

The greater response of the high-Myc cells to stimuli is exactly what would be expected if Myc is a universal transcription amplifier. We now note this on **Page 12**.

The use of HDAC inhibitors was not helpful. Is FUBP1 level/binding to the Myc region different in the two populations, +/-PDGF?

This is now shown in **Figure 6B** and **Supplemental Figure 7D** and discussed on **Page 13**.

Figure 5 reports the broad FUBP1 binding to the first exon/intron and to the 3' UTR region of the mouse Myc gene by ChIPSeq experiments. Do the extent/sites of FUBP1 binding change in the low versus high-Myc populations and in response to PDGF/TNF- α ?

We now show FUBP1's binding at the *MYC* locus in the high vs low MYC cells in **Supplemental Figure 7E** and discussed on **Page 14**. And we now show FUBP1's binding at the MYC locus in response to PDGF/TNF- α in **Figure 6B** and discussed on **Pages 14-16**.

Response to Reviewer 3

Reviewer #3 (Remarks to the Author):

General comments

This manuscript questions the robustness and the range of MYC expression variation, at single cell level, in primary cells. The authors were able to separate two cell fractions of a MEF population, each of them characterized by their MYC expression level (low and high). They demonstrated that the low-MYC sub-population do not display MYC expression modulation upon stimulation. The high-MYC sub-population displayed an increase in MYC expression upon strong stimulation. Loss of combined FUBP1 and FUBP2 maximize this increase upon stimulation of the total cell population. On the contrary, the high-MYC sub-population showed a decrease in MYC expression upon exposure to DNA methylation agent and/or HDAC inhibitor, demonstrating an epigenetic regulation. The approaches are appropriate and the data are convincing. Still, the manuscript shows some weaknesses in some demonstrations and will gain impact if the authors were able to demonstrate the (direct or indirect) causative link between epigenetics modification and FUBP1/2 function(s).

Review

Major points :

-The authors have to justify more the use of MEF cells and naïve B cells for their study. In particular, they have to include references for data reporting what is known on those lineages for myc or fubp1/2, if those data exist. If those data do not exist, the authors have to present data on the level of expression of FUBP1 and FUBP2 on those primary cells at steady-state level, as well as upon stimulation and treatment (aza and TSA).

MYC perturbation is so pervasive in cell-lines, that we were compelled to use primary cells that had not experienced manipulations or culture conditions that might sponsor oncogenic or pre-oncogenic changes. Studies of the variation in expression of MYC during short-term B-cell activation used fresh murine B-cells. Due to embryonic lethality, FUBP^{-/-} B-cells were not available, and so to explore the influence of the FUBPs on steady-state and activated MYC-expression and to monitor MYC-levels through sequential passages, low passage wild-type and FUBP^{-/-} MEFs were employed. We now include data that shows the FUBP1/2 levels as requested in MEFs before and after growth factor/cytokine treatment.

- The authors have to justify the choice of the low and high concentration of FGF, IL4, PDGF, TNF α . What was the rationale based on? For example, what are the known associated functions activated or inhibited at those concentrations?

- Explain also why low IL4 or low FGF is considered physiological and high FGF or IL4 pathological. Provide references, elaborate more on this point or reformulate.

In the preparatory experiments leading to the work presented in this manuscript, we performed cytokine and growth-factor curves to define the concentrations that yield the maximal-MYC induction and maximal growth using low passage primary MEFs. At the concentrations employed apoptosis was not observed. The factors used, IL-4, PDGF, FGF, and TNF α , are all induced by stress, inflammation, infection, and/or neoplasia—all pathologic processes. We do not use the term “pathologic” to assert that there is pathologic over-production of these factors, but rather that the levels that sponsor maximal MYC expression are beyond the range associated with unstressed tissue homeostasis. We now *operationally* define physiological as a weak/homeostatic response and “stressed/pathological” as an intense/maximal response to a major perturbation. The insight of reviewer 3 is appreciated as a review of the literature indeed complicates a strict definition of “normal” versus “pathologic”. Normal and pathologic are defined using a bewildering array of different experimental systems, cell-lines, diseases, and assays for cytokine and growth factor levels and function. However common to all of them (and we spent a lot of time scouring the literature) is that the statistical separation between normal versus pathologic typically occurs over a growth-factor/cytokine concentration

range of two to four-fold—the same sort of range that we empirically use for IL-4, TNF α , PDGF and FGF. (It should be noted that in some human tumors, FGF levels may become elevated more than 10x). Notably, the same concentrations of PDGF and TNF α concentrations that we employed separate normal and pathologic levels in at least one study (Andrade Junior *et al.*, 2008; Koizumi *et al.*, 2015). There is a large literature that associates higher level cytokine, growth factor or lymphokine concentrations with a variety of pathologic conditions. The dose selected for low-level stimulation with cytokines and growth factors that define stage I, leaves enough room for upregulation to maximal levels in stage II.

- Very importantly, a clarification must be provided for the definition of stage I, stage II and MYC setpoint. Stage I and stage II are concepts that apply to a cell population since the stage I-stage II boundary is defined in the cumulative distributive fraction of the cell population. How do the authors integrate those concepts at a single cell level? Are the author able to translate the stage I-stage II boundary into a clear threshold in Figure 3D (protein level) for instance? The authors have to rephrase the manuscript to redefine/clarify the stage I-II boundary (cell population) or threshold (single cell). If there is no difference, use the same word. Clarify the definition of MYC-setpoint compared to the stage I-II boundary.

We appreciate the reviewer's point. In fact, we take snapshots of cell populations but do not track individual cells across time, and so at the reviewer's suggestion, we have edited the manuscript to use the term boundary almost exclusively (we retain one usage in the Discussion).

Associated to Figure 3, it is stated "... to reduce the range of MYC expression in Stage II ». This statement is not appropriately quantified or demonstrated. In fig 3D, for instance, the range of MYC expression does not seem a lot different between WT and KOKD conditions. To me, the range of MYC expression looks the same, it is the number of cells expressing high-MYC that seem to be different. Could the author make the manuscript more clear on this point ? We have added **Figure 5E** to better report these results.

- It would be interesting to show the proliferation status of KD, KO and KOKD cells compared to WT cells, upon treatment and non treated cells in order to draw correlation between the level MYC, level of FUBP1/2 and the proliferation rate. At least two publications (Rabenhorst, Cell Report 2015 and Debaize, Nucleic Acid Research, 2018) demonstrate that FUBP1 sustain proliferation (in mouse or human).

The proliferative status of the WT and KO are now compared in **Supplemental Figure 5C**, and shows comparable results. Note that the KOKD cells are not viable after ~ 5-days.

- Data from Rabenhorst, Cell Report 2015 show that FUBP1 shRNA decreases the level of Myc mRNA. How the

authors also articulate their results with the report showing that FUBP1 alone is not sufficient to activate MYC expression, but it is required for its maximal activation (He L, Liu J, Collins I et al (2000) Loss of FBP function arrests cellular proliferation and extinguishes c-myc expression. EMBO J 19:1034–1044) ?Could the authors comment/discuss more on those apparently contradictory data with their Fig 3A and E ?

The influence of FUBP1 on the expression of MYC, p21 and indeed on cancer where it is prooncogenic *and* a tumor suppressor, is both positive and negative. In previous studies, we have provided evidence in support of the hypothesis that FUBP1 is a molecular cruise-control that dampens both upward and downward fluctuations in MYC levels. Just as a cruise-control must actuate both the accelerator and the brake, depending on circumstances, so FUBP1 can either up- or downregulate transcription. (This is now discussed on **Page 20** of the revised manuscript.) This may also account for FUBP1 paradoxical roles in cancer as introduced on **Page 4** of the revision.

To estimate the specificity of the binding of FUBP1 on the Myc locus, it would be informative to show chip-seq data on other genes known to be regulated by FUBP1/2 and genes that are not. Indeed, what is striking here is the broad FUBP1 signal in the myc locus. Is FUBP1 bound everywhere like this in the genome? Can the authors provide some whole-genome analysis of their Fubp1-Chip-Seq and comment on it? The authors could also perform Chip-Seq (or Chip-qPCR) on the myc locus in MEF KD, KO and KOKD cells to demonstrate the implication of FUBP1/2 binding at those regions on the genome and MYC expression variability. The same after TNF or PDGF stimulation.

Some of the requested information is shown in **Supplemental Figures 6 and 7B**, and **Figure 6B**.

- The causative link between FUBP1/2 and epigenetically-controlled Myc-setpoint is not clearly demonstrated (see the title). For example, are the level of H3K27ac, H3K4me1, H3K4me3 modified at this locus in MEF KD, KO and KOKD cells? or, is the level of MYC sensitive to aza or TSA in MEF KD, KO and KOKD?

We have examined the level of H3K4me1, H3K4me3 and H3K27ac at *Myc* locus in MEF KD, KO and KOKD cells using ChIP-qPCR and now report the results in **Figures 6B and 6C**, and **Supplemental Figures 7E and 8**.

-The text and data are minimalists. Important information is lacking. Among them:

- (1) The numbers of replicates must be indicated for each figure. Please clearly state whether or not all the graphs with 'fraction of cells' represent one experiment or are the sum of many (and indicated the number).
- (2)The following data are required and must be displayed in supplemental figures: western blots allowing estimating the efficacy of extinction of Fubp1 and Fubp2 in KD and KO.
- (3) Show the density curves corresponding to Fig2B and C, in supplemental data (those representation are more usual), corresponding to all the conditions (negative, untreated, low and high). Doing so, the 'peak value' will be clear.

Indeed, the reviewer is correct. Showing the density plots of the IL-4 treated B-cells and the FGF treated MEFs in **Supplemental Figure 2** (and reported on **Page 8** of the revised text) nicely illustrates that mode-value of MYC shifts upwards upon modest stimulation expression does not enter stage II without intense stimulation. We thank the reviewer for this suggestion.

(4) Statistics should be done and indicated when appropriate on each figure.

We have added the replicates and number of experiments and statistics to the figures.

(5) Provide sufficient information (or appropriate references) on CHIP-seq in the material and method so that the experiment can be repeated by other labs. Provide also the external link to the deposit of whole CHIP-Seq data.

We have added this information to the Materials and Methods on **Page 25**.

(6) The characterization of MEF myc-egfp cells or mice needs to be provided, including the homozygotic status.

This has been shown on **Pages 8-9**.

Minor points:

- In the abstract: the causative link between FUBPs and epigenetic setpoints has not been fully demonstrated. Moreover, in absence of stimulation, it seems to be exaggerated to write “Cells lacking ssDNA-binding proteins FUBP1 and FUBP2 overpopulate stage II “. Thus these two sentences should be edited. « Cells lacking ssDNA-binding proteins FUBP1 and FUBP2 overpopulate stage II, even when unstimulated. Thus, the FUBPs help to enforce constraints on the epigenetic setpoints that restrict MYC expression to stage I. ».

- B naive cells were used only in Figure 2, and not in the rest of the manuscript. This result is not of major interest for this manuscript. I suggest transferring those B-cell data in the supplemental part of the manuscript. Data on RNA level of MYC in naïve B cells are also lacking.

- justify the change of compound used to stimulate the cells since FGF is no longer used in MEF cells.

FGF was used and similar results were obtained. We’ve added the CDF curve and box plot in **Supplemental Figures 5B** and **5E**, to compare the steady-state and FGF-treated samples.

- In the legend of Fig3, the concentration of PDGF and TNF must be indicated.

Thank you. We’ve added this information.

- in the discussion, the authors can comment on plausible direct MYC-target genes among the list on modulated genes (supplemental Table 1).

MYC targets almost all active genes (Nie Z. et al. Cell 2012; Lin, CY et al. Cell 2012).

- The authors may also discuss on another role of FUBP1/2 that could be implicated in the control of the MYC-setpoint, by participating in different promoter-enhancer interaction. (and where some enhancer can be accessible or not by DNA methylation or Histone modification).

Data relating to chromatin modifications have been added.

REVIEWERS' COMMENTS:

Reviewer #1 (Remarks to the Author):

The manuscript by Zheng and colleagues is strongly improved after revision and is, in my opinion, suitable for publication.

Reviewer #2 (Remarks to the Author):

The paper by Zheng et al., entitled: "FUBP1 and FUBP2 Enforce Distinct Epigenetic Setpoints for MYC Expression in Primary Single Cells" reports the results of a study concerning the level of Myc expression in single mouse embryo fibroblasts (MEFs).

In the revised version, the authors made an effort to meet the Reviewer's concerns by including more controls and new experiments/panels in several figures. Most answers are satisfying. However, some concerns still hold:

1. Reviewer's comment: Despite the wealth of data, this paper is rather descriptive, with minimal mechanistic information. Little investigation is directed to the mechanism underlying the interesting bimodal Myc expression in the MEF population and the mechanistic relevance of the FUBP1/2 regulators to the Myc setpoints. Also, no explanation/model are provided as to why the extent of PDGF- or TNF- α -dependent induction is higher in the high- than in the low-Myc expressing population.

Response: We have made numerous changes to the text and figures to address the reviewer's critique. We have identified chromatin features that distinguish high- versus low-MYC and FUBP1 wild-type versus KOKD MEFs. This has been added to Figure 6 and is discussed on Page 15. We now consider that increased MYC-driven transcription amplification, especially of inflammation/immunity genes in high-MYC cells may enable these cells to act as "sentinels" for stress and pathologic processes. (Page 11).

Reviewer to revised version: It is unclear which are the high and low Myc samples in Fig 6, please mark them clearly. By including chromatin studies, the authors further characterise in a descriptive manner the sorted high and low Myc cell populations, besides the previously described differences in high expression of inflammation/immunity genes, high proliferation rate, and PDGF- or TNF- α -dependent induction in the high-Myc cell population.

2. Reviewer's comment: In Figure 4, the two high and low Myc expressing populations are sorted and shown to maintain their respective Myc expression levels (4b and c). Both populations are characterised for their proliferation rate (4d), inflammatory and immune response gene expression (4e).

However, the authors did not test the two populations for the expression of FBP-1 and 2: if these DNA-binding factors are relevant to the Myc expression, they may level differently in the high versus low Myc population.

Response: The mRNA-seq on the low- and high-MYC cells shows there is a significant increase in gene expression in high-MYC cells compared to low-MYC cells and almost no downregulated genes. In parallel with the increase of MYC, Fubp1 and Fubp2/Khsrp mRNA in high-MYC cells increased about 2 times than those in low-MYC cells (Figure for the Reviewers 1).

Reviewer to revised version: Although the effect is not great, "Figure for the Reviewers 1" should be shown in the revised manuscript.

Minor points:

Figure legends (main and supplementary) should be accurately checked, to consider the new

panels included in the revised version.

Reviewer #3 (Remarks to the Author):

The authors have addressed the comments of the reviewers in a satisfactory and relevant way. Additional important experiments have been performed to enforce the message on epigenetic changes. As a result, the manuscript has greatly improved and gained in clarity and depth.

The following points must be addressed before publication:

1/In the abstract: "Growing cells remember low and high MYC setpoints through multiple cell divisions but are limited by the same expression ceiling even after modest MYC- activation (Stage I). » should read « Growing cells remember low and high MYC setpoints through multiple cell divisions and are limited by the same expression ceiling even after modest MYC- activation (Stage I). »

2/In the abstract: "Lacking FUBPs, even unstimulated cells populate stage II, and sponsor MYC chromatin changes » should read « : "Lacking FUBPs, even unstimulated cells populate stage II, and sponsor MYC chromatin changes, revealed by histone marks»

3/ When the authors refer to Z-DNA, please give a definition or brief description of what it is.(Something like: Z-DNA, a specific double helical structures of DNA).

4/ In the discussion: the following sentence should read: "FUBP1 has been found to associate with a variety of genes that regulate cell growth or death besides MYC including p21, Ccna1, USP29, c-KIT through binding to FUSE-like upstream regulatory sequences." Relevant references must be added including Rabenhorst, Cell Report 2015 and Debaize et al, NAR 2018.

Finally, the following point is suggested and should be taken in consideration if it is in the editorial recommendations.

5/Provide the external link to the deposit of whole ChIP-Seq data. (eg : GEO repository for instance).

REVIEWER 3

The authors have addressed the comments of the reviewers in a satisfactory and relevant way. Additional important experiments have been performed to enforce the message on epigenetic changes. As a result, the manuscript has greatly improved and gained in clarity and depth.

The following points must be addressed before publication:

1/In the abstract: "Growing cells remember low and high MYC setpoints through multiple cell divisions ~~but~~ are limited by the same expression ceiling even after modest MYC-activation (Stage I)." » should read « Growing cells remember low and high MYC setpoints through multiple cell divisions and are limited by the same expression ceiling even after modest MYC- activation (Stage I). »

2/In the abstract: "Lacking FUBPs, even unstimulated cells populate stage II, and sponsor MYC chromatin changes » should read « : "Lacking FUBPs, even unstimulated cells populate stage II, and sponsor MYC chromatin changes, revealed by histone marks»

3/ When the authors refer to Z-DNA, please give a definition or brief description of what it is. (Something like: Z-DNA, a specific double helical structures of DNA).

4/ In the discussion: the following sentence should read: "FUBP1 has been found to associate with a variety of genes that regulate cell growth or death besides MYC including p21, Ccna1, USP29, c-KIT through binding to FUSE-like upstream regulatory sequences." Relevant references must be added including Rabenhorst, Cell Report 2015 and Debaize et al, NAR 2018.

Finally, the following point is suggested and should be taken in consideration if it is in the editorial recommendations.

5/Provide the external link to the deposit of whole CHIP-Seq data. (eg : GEO repository for instance).

Response to reviewers' minor comments

Reviewer 1: No comments.

Reviewer 2:

1. **It is unclear which are the high and low Myc samples in Fig 6, please mark them clearly.** RESPONSE: Figure 6 shows neither high or low MYC samples, it compares FUBP1 binding and chromatin features in cells manipulated for FUBP1 and FUBP2 expression. These cells were not sorted according to their MYC expression level.
2. **Although the effect is not great, "Figure for the Reviewers 1" should be shown in the revised manuscript.** RESPONSE: The "figure for reviewers" has been included in Supplementary Figure 4.

Reviewer 3: 1/In the abstract:" Growing cells remember low and high MYC setpoints through multiple cell divisions but are limited by the same expression ceiling even after modest MYC-activation (Stage I). » should read « Growing cells remember low and high MYC setpoints through multiple cell divisions and are limited by the same expression ceiling even after modest MYC- activation (Stage I). »

2/In the abstract: "Lacking FUBPs, even unstimulated cells populate stage II, and sponsor MYC chromatin changes » should read « : "Lacking FUBPs, even unstimulated cells populate stage II, and sponsor MYC chromatin changes, revealed by histone marks»

3/ When the authors refer to Z-DNA, please give a definition or brief description of what it is.(Something like: Z-DNA, a specific double helical structures of DNA).

4/ In the discussion: the following sentence should read: "FUBP1 has been found to associate with a variety of genes that regulate cell growth or death besides MYC including p21, Ccna1, USP29, c-KIT through binding to FUSE-like upstream regulatory sequences." Relevant references must be added including Rabenhorst, Cell Report 2015 and Debaize et al, NAR 2018.

5/Provide the external link to the deposit of whole ChIP-Seq data. (eg : GEO repository for instance).

RESPONSES: All of the reviewer's comments have been addressed exactly as requested.